# Rapid, label-free histopathological diagnosis of liver cancer based on Raman spectroscopy and deep learning

Liping Huang [1,2], Hongwei Sun[3], Liangbin Sun[1], Keqing Shi[3], Yuzhe Chen[1], Xueqian Ren[1], Yuancai Ge[1], Danfeng Jiang[2], Xiaohu Liu[1], Wolfgang Knoll[4], Qingwen Zhang [2] ✉ & Yi Wang [1,2] ✉

Biopsy is the recommended standard for pathological diagnosis of liver carcinoma. However, this method usually requires sectioning and staining, and well-trained pathologists to interpret tissue images. Here, we utilize Raman spectroscopy to study human hepatic tissue samples, developing and validating a workflow for in vitro and intraoperative pathological diagnosis of liver cancer. We distinguish carcinoma tissues from adjacent non-tumour tissues in a rapid, non-disruptive, and label-free manner by using Raman spectroscopy combined with deep learning, which is validated by tissue metabolomics. This technique allows for detailed pathological identification of the cancer tissues, including subtype, differentiation grade, and tumour stage. 2D/3D Raman images of unprocessed human tissue slices with submicrometric resolution are also acquired based on visualization of molecular composition, which could assist in tumour boundary recognition and clinicopathologic diagnosis. Lastly, the potential for a portable handheld Raman system is illustrated during surgery for real-time intraoperative human liver cancer diagnosis.

Liver cancer was the seventh most frequent cancer and the third leading cause of cancer-related death worldwide in 2020[1]. The incidence of newly diagnosed cases and age-standardized incidence rates of liver cancer have continued to increase globally in the past few decades, despite significant advances in diagnosis and therapy[2,3].

Therefore, accurate and timely diagnosis is crucial for the treatment of liver cancer and the improvement of the survival rate. Serological testing combined with imaging is the standard method for the diagnosis of hepatic carcinoma[4]. However, the diagnostic sensitivity of the most commonly used serological test, which assays for alpha-fetoprotein (AFP), is ~60%[5]. Imaging tests such as magnetic resonance imaging (MRI), computed tomography (CT), and ultrasonography (US) have high sensitivity and specificity for liver cancer detection, especially in patients with liver cirrhosis[6]. Such imaging tests, however,

suffer from limited spatial resolution, complexity on intraoperative diagnosis and/or carry a risk of exposure to ionizing radiation[7]. Therefore, the biopsy is still recommended as a gold standard for pathological diagnosis, which is important for prognosis and guiding treatment[4].

Clinically, histopathological observations are usually carried out with haematoxylin & eosin (H&E) or immunohistochemical staining. The staining procedure is time-consuming and is only suitable for diagnosis using isolated tissues. Additionally, the limited number of pathology specialists may restrict the use of histopathology[6]. Recently, digital pathology using high-throughput image analysis has greatly assisted pathologists in tissue sample identification and classification[8,9]. However, sample preparation for digital pathology suffers from the same limitations as traditional methods. Therefore,

[1]School of Biomedical Engineering, School of Ophthalmology and Optometry, Eye Hospital, Wenzhou Medical University, 325001 Wenzhou, PR China. [2]Engineering Research Center of Clinical Functional Materials and Diagnosis & Treatment Devices of Zhejiang Province, Wenzhou Institute, University of Chinese Academy of Sciences, 325001 Wenzhou, PR China. [3]The First Affiliated Hospital of Wenzhou Medical University, 325015 Wenzhou, PR China. [4]Austrian Institute of Technology, Giefinggasse 4, Vienna 1210, Austria. ✉e-mail: zhangqw@wiucas.ac.cn; yiwang@wmu.edu.cn

techniques are needed for faster and non-disruptive in vitro and even in vivo investigation of liver cancer.

Spectral histopathology based on Raman spectroscopy provides an alternative approach to cancer diagnosis[10]. Raman spectroscopy is an optical technique based on the inelastic scattering of light by vibrating molecules that provides chemical fingerprints of complex biological samples, and most biomolecular information is available with just a simple snapshot with the Raman measurement. Importantly, the chemical structure and composition of biological samples can be obtained by Raman spectroscopy in a stain-free and non-destructive manner with minimal sample preparation[11–14]. Spectral information can also be combined with artificial intelligence algorithms to establish a diagnostic classification model to allow automatic diagnosis[15–18]. Furthermore, Raman spectroscopy imaging allows for the delineation of tumour margins and visualization of lesion regions of interest that are invisible to the naked eye[19]. These features make Raman spectroscopy feasible for the examination of isolated tissue specimens and the assistance of surgeons to identify the margins of tumours, facilitating more complete removal with minimal damage to normal tissue.

Thus far, there have been investigations of the use of Raman spectroscopy for pathological diagnosis of several biological tissues, including brain[20], breast[21], skin[22,23], colon[24], and bladder[25]. For liver cancer, studies based on Raman spectroscopy have primarily focused on the analysis of blood samples, with only a few studies aimed at human tissue.

In addition, it is known that the heterogeneity of tumour tissues and possible carcinoma infiltration increase the variability of spectral data collected from tissues. Therefore, it is necessary to collect a large number of spectra from each tissue sample to better represent the data, but this may increase the complexity of data analysis and pose a challenge for traditional chemometric methods. The data-driven nature of deep learning is well suited to solve this problem[17]. Deep learning can extract and learn hidden features directly from massive data and has been successfully applied in the field of image recognition, including biological and medical image analysis[26–28]. Thanks to the flexibility of its architecture, deep learning has also been extended to analyse one-dimensional sequential data, such as spectral data[29,30]. A few reports described deep learning for medical diagnosis using 1-D Raman spectral data[18,31].

In this study, we reported exploration of human hepatopathy tissue using Raman spectroscopy. We first successfully distinguished hepatic carcinoma tissues from adjacent non-tumour tissues using Raman spectroscopy combined with a VGG-16-based convolutional neural network (CNN), in a rapid, non-disruptive, and label-free manner. A more detailed pathological identification was then made of liver cancer tissues, including subtype, differentiation grade, and tumour stage. Tissue metabolomics analysis confirmed the reliability of Raman spectroscopy in the identification of metabolites. Furthermore, Raman images of unprocessed human tissue blocks and tissue slices at sub-micrometric resolution allowed visualization of their molecular composition, facilitating the identification of tumour boundaries and clinicopathologic diagnosis. Finally, a handheld Raman spectroscopy system was employed during surgery to explore the feasibility of real-time intraoperative liver cancer diagnosis. A graphical workflow of the liver tissue histopathological diagnosis and intraoperative diagnosis based on Raman spectroscopy and an intelligent algorithm is shown in Fig. 1.

## Results

### Raman analysis of liver tissues

An excitation wavelength of 532 nm was used for the Raman measurements reported here, whereas longer wavelengths are usually recommended for biological sample analysis to avoid fluorescence background signals and obtain deeper light penetration. However, compared with longer wavelengths (such as 633 or 785 nm, respectively), the shorter wavelengths provided higher data quality and signal-to-noise ratio for Raman spectra (Supplementary Fig. 1), which partly resulted from resonant amplification of specific protein and carotenoid-associated bands[32].

Raman spectra of matched hepatic carcinoma tissues and adjacent non-tumour tissues were acquired from 120 liver cancer patients. Detailed patient information is listed in Supplementary Table 1. Due to the heterogeneity and complexity of cancer tissues (Supplementary Fig. 2), at least 50 spectra were collected from randomly selected points on the surface of each tissue sample. A comparison of average Raman spectra obtained from carcinoma and paracarcinoma tissue samples is shown in Fig. 2a. Nineteen characteristic Raman peaks were observed from most tissue samples. The Raman peaks of the two groups largely overlapped, but the intensity of each peak in the paracarcinoma tissue group is significantly higher than that in the cancer tissue group (Student's $t$-test, $P < 0.05$). Hierarchically clustered heatmaps of characteristic Raman peaks were plotted to pre-discriminate the closely related Raman peaks (Supplementary Fig. 3). Supplementary Table 2 gives the peak position and corresponding representative compounds of the major Raman vibrational modes reported in the literature[33].

Most of the peaks could be ascribed to aromatic amino acids, proteins, and carotenoids. Specifically, the peaks at around 749, 1212, 1393, 1547, 1586, and 1602 cm$^{-1}$ are related to tryptophan, tyrosine, or phenylalanine, respectively. The band at 1637 cm$^{-1}$ corresponds to the C = O stretching of the amide I band. The Raman lines at 1003, 1156, and 1519 cm$^{-1}$ represent the C−C and C−N stretch of carotenoids. The peak at 1003 cm$^{-1}$ has also been reported to be associated with AFP, a biomarker for hepatocellular carcinoma (HCC)[34]. Furthermore, the Raman features appearing at 1081, 1130, and 1304 cm$^{-1}$ are mainly related to lipids or fatty acids. The bands at 674, 974, 1336, and 1356 cm$^{-1}$ can be assigned to nucleic acids. Additionally, the Raman signature around 835 cm$^{-1}$ is related to saccharides. Such differences in Raman spectra between paracancer and cancerous tissue reflect variations in biochemical components of liver tissue caused by carcinogenesis, which provides a basis for differentiating cancerous from normal tissues.

Next, fine histopathological differentiation of liver cancer based on Raman spectroscopy was further explored. Primary liver cancer is one of the most common cancers worldwide, of which 75–85% and 10–15% are hepatocellular carcinoma (HCC) and intrahepatic cholangiocarcinoma (ICC), respectively[35]. As shown in Fig. 2b, the major differences in Raman signal intensity between the HCC and ICC groups were at carotenoid-related Raman peaks (1003, 1156, and 1519 cm$^{-1}$), which were significantly higher in the ICC group. In contrast, most amino acid, lipid, and nucleic acid-related peaks, such as 749, 974, 1304, 1356, 1393, and 1586 cm$^{-1}$, respectively, were higher in the HCC group. In addition, an accurate judgment of tumour stage and differentiation grade may assist in the choice of treatment strategy and prognosis assessment. As shown in Fig. 2c, the primary spectral differences between early and advanced stages were found at 1003, 1156, 1519 cm$^{-1}$ (carotenoids), 1130 cm$^{-1}$ (fatty acids), 749 and 1547 cm$^{-1}$ (tryptophan), which had higher intensity in the early-stage group, while the peaks related to nucleic acids at 674 and 974 cm$^{-1}$, and saccharide at 835 cm$^{-1}$ were higher in the advanced stage group. Furthermore, Supplementary Fig. 4 shows the Raman spectra of different categories of cancer differentiation. The overall spectral difference between moderately and poorly differentiated groups was more significant than that of well- and moderately differentiated groups. Similar to the differences between cancer and paracancer groups, well- and moderately differentiated groups also exhibited a higher overall spectral intensity, especially in the carotenoid-related peaks (Fig. 2d). The pathologic type of each tissue block was reconfirmed by the pathologist based on H&E staining after the Raman testing (Fig. 2e–g).

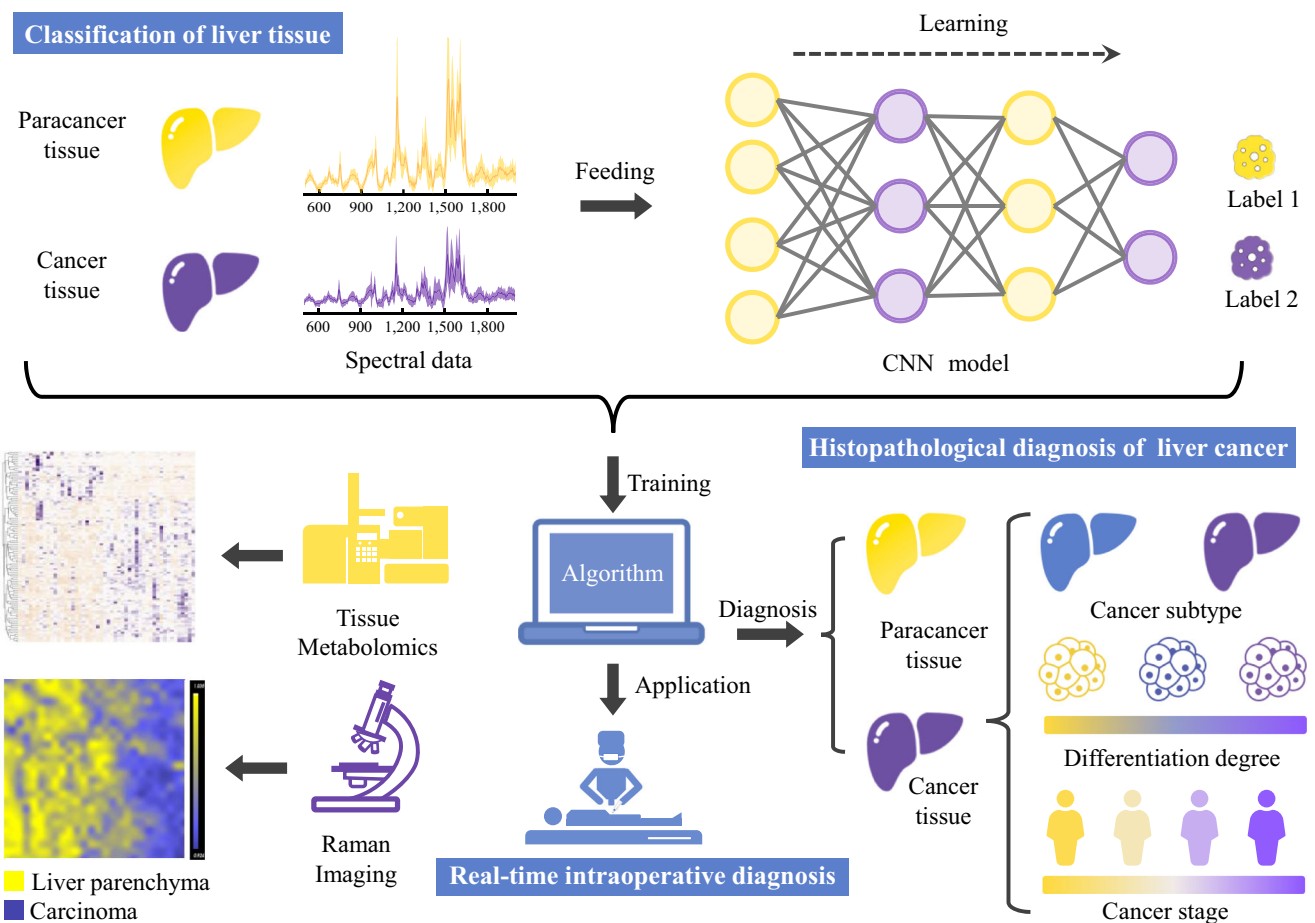

**Fig. 1 | Workflow of histopathological diagnosis of liver cancer based on Raman spectroscopy and an intelligent algorithm.** Large Raman datasets acquired from liver tissue were collected and fed into a CNN-based deep learning model to train it to distinguish spectral data of different tissue types. The model was then used to differentiate different pathological types of liver cancer tissues. In addition, the Raman results were validated by tissue metabolomics based on liquid chromatography–mass spectrometry (LC–MS). Furthermore, Raman images were used to visualize the molecular composition of unprocessed human tissue blocks and tissue slices. Finally, a handheld Raman system was employed during surgery for real-time intraoperative liver cancer diagnosis.

## Histopathological diagnosis of liver tissues based on deep learning

To classify different types of liver tissues using Raman spectra, a VGG-16 network-based CNN model was employed. The model architecture consisted of 13 one-dimensional convolutional layers, 5 pooling layers, and 3 fully connected layers (as shown in Fig. 2h), utilizing small-scale convolution kernel stacking rather than large-scale convolution kernels to reduce the parameters required for calculations[36]. A liver tissue Raman database was established with 50 spectra per tissue sample, and a total of 12,000 spectra were obtained from 120 pairs of liver tissue samples. The spectral data ranged from 500 to 2000 cm$^{-1}$ with 889 one-dimensional float data. A binary classification model was built to classify liver cancer tissue and paracancer tissue, which were designated 1 and 0, respectively. The spectral data were pre-processed with baseline subtraction and smoothing and then fed into the CNN model with a random shuffle. The softmax function was used as the activation function in the output layer, which output probabilities of two classes with the higher value considered to be the predicted class.

Accuracy and cross-entropy loss are two indicators that are often used to assess the performance and reliability of CNN models. With learning iterations, the accuracy and cross-entropy loss curves of the validation set gradually tend to converge, indicating that the model is not over-fitting (Fig. 2i, j). As a result, an accuracy of 92.6% was obtained for estimating carcinoma tissue area, accompanied by a sensitivity and specificity of 90.8% and 94.6%, respectively.

In addition, three other CNN models were established to distinguish HCC from ICC tissues, and among tissues with different cancer stages and differentiation grades. The performance of four binary models is shown in the confusion matrices of Fig. 2k. Tumour heterogeneity posed a challenge in the discrimination of different stages and differentiation grades of tumour tissues, with accuracies of 78.3% and 72.3%, respectively. But a better result was acquired for the classification of the liver cancer subtypes HCC and ICC, yielding an identification accuracy of 82.4%. Four receiver operating characteristic (ROC) curves were plotted to quantitatively verify the performance of classifiers (Fig. 2l), with the area under the curve (AUC) values between 0.783 and 0.965. Furthermore, compared with other common machine learning algorithms, including PLS-DA, random forest, and XGBoost, the deep learning approach shows superior computational performance with higher accuracy in tissue identification of different pathological types, especially in dealing with imbalanced data (Supplementary Table 3).

It is worth noting that conventional diagnosis of HCC based on a single serological biomarker (such as AFP) achieved poor sensitivity in this study. At an AFP threshold of 200 ng/ml, 25 of 92 HCC patients were positive, with a sensitivity of only 27.2% (Supplementary Fig. 5), much lower than our method based on Raman measurements. Besides, imaging modalities such as CT and MRI are recommended as the firstline diagnostic methods to identify or predict different pathological states of HCC[37,38]. For example, clinical staging of HCC is

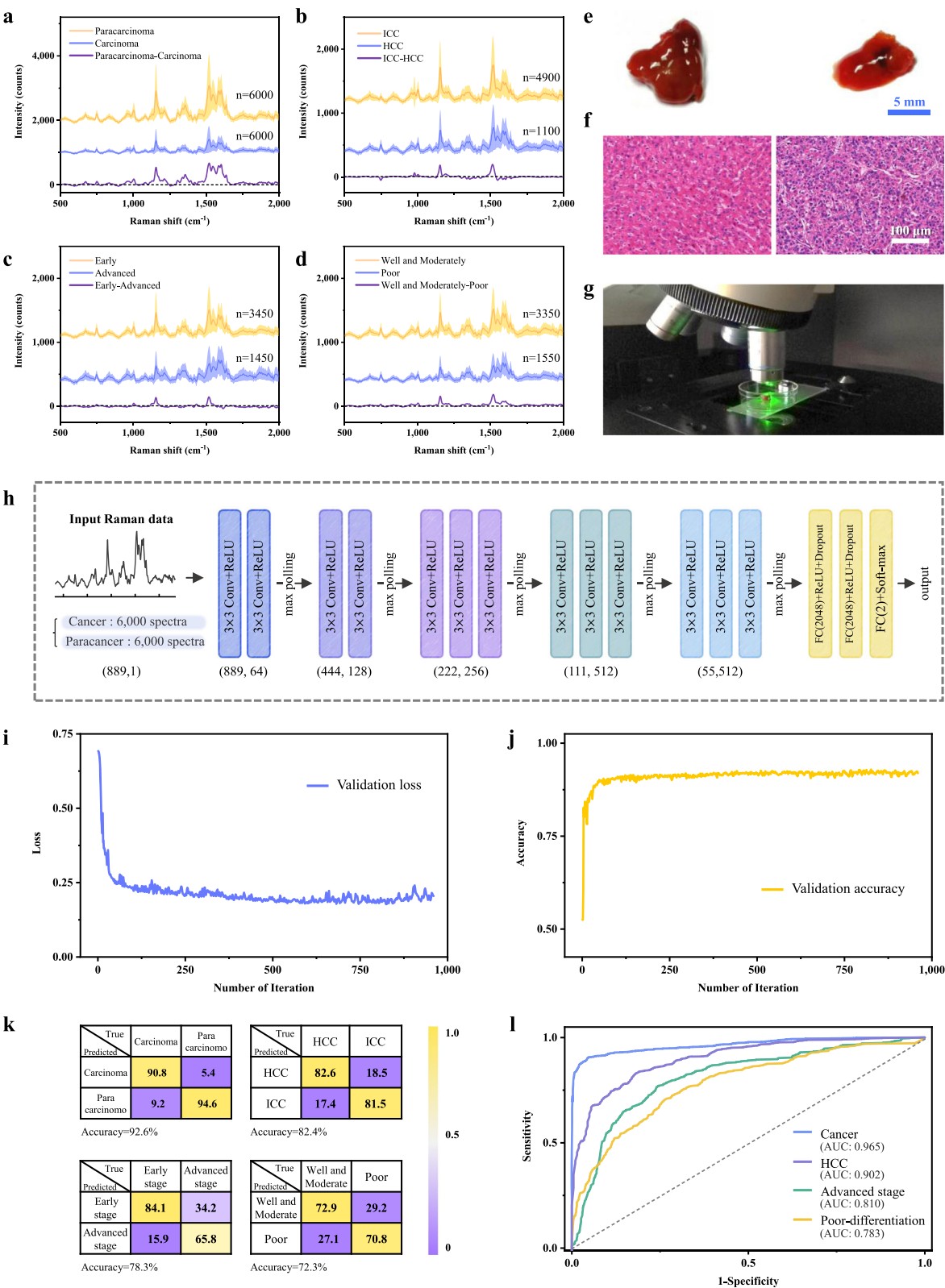

mainly diagnosed based on imaging characteristics, including the number and size of HCC nodules, and the presence of vascular invasion. Here, Raman spectra have also shown feasibility for the determination of microvascular invasion with an accuracy of 67% and an AUC value of 0.694 based on 84 patients (Supplementary Fig. 6). The results may be improved with a further increase in sample numbers and spectral collections. In sum, in terms of accuracy, Raman spectroscopy is comparable to or better than traditional imaging modalities (such as CT, MRI, and US) in identifying different pathology types in the current study (Supplementary Table 4), providing a powerful complement to existing pathology diagnostic techniques.

**Fig. 2 | Raman analysis of liver tissues and histopathological diagnosis based on deep learning. a–d** The average Raman spectra of 120 carcinoma and 120 para-carcinoma tissue samples (**a**), cancer tissue samples of patients with HCC and ICC (**b**), HCC tissue samples at different tumour stages (**c**), and HCC tissue samples with different cancer cell differentiation grades (**d**). The shaded areas represent the standard deviations of the mean. **e** and **f** Typical photographs of paracancer tissue (left) and liver cancer tissue sample (right) (**e**) and the corresponding images of the H&E-stained tissues (**f**) from 120 tested samples in this study. **g** Raman testing of liver tissue with a micro-Raman spectrometer. **h** The architecture of the VGG-16-based deep learning model. Raman data consisting of 12,000 spectra were fed into the initial convolutional layer with 64 filters. Each convolutional layer had a kernel size of 3, connecting with a ReLU activation layer. A drop-out layer was utilized in full connection layers, following the basic blocks. Max-pooling (size 2, stride 2) was employed between blocks to reduce data length. The numbers below each block refer to the length and the number of channels of output respectively. **i**, **j** Cross-entropy loss (**i**) and accuracy (**j**) in iterative training of the CNN. Cross entropy represents the mean square error between the predicted value and the true value. **k** Binary confusion matrices for the classification of four tissue categories based on the CNN algorithm in percent (%). **l** ROC curves and corresponding AUC values. Source data are provided as a Source Data file.

## Tissue metabolomics and serological analyses

In order to further confirm changes in the biochemical composition of liver cancer tissues, a non-targeted metabolomics strategy was employed based on liquid chromatography–mass spectrometry (LC–MS). Tissue metabolomics is widely used in the study of disease pathogenesis based on metabolic characteristics[39,40], which can provide direct information about metabolic alterations at targeted sites and reveal relevant tumour biomarkers. A total of 25 pairs of matched HCC tissues and adjacent non-tumour tissues were assessed in this study. 1995 and 2228 ions were retained in positive and negative electrospray ionization source (ESI$^+$ and ESI$^-$) modes, respectively (Supplementary Data 1), after removing deviation and missing values. In total, 57 metabolites in ESI$^+$ mode and 51 in ESI$^-$ mode were identified and selected as candidate differential metabolites (Supplementary Data 2). The differences between nine types of primary metabolites and the hierarchically clustered heatmap of 108 specific metabolic biomarkers between HCC tissues and adjacent non-tumour tissues were plotted in Fig. 3a and b. Most of the metabolites exhibited a downward trend in HCC tissues, which is consistent with the lower Raman intensity in HCC tissues.

For instance, a significant down-regulation of tyrosine was observed in HCC tissues, while the other two aromatic amino acids (ArAAs), phenylalanine and tryptophan, did not have significant changes (Fig. 3a(i) and Fig. 3c). However, we found decreases of other aromatic compounds in HCC tissues, including dopa, 3-hydroxyanthranilic acid, and aniline, suggesting that variations in benzene-ring related Raman bands from liver tissues might also derive from ArAA derivatives or other aromatic metabolites. In addition, most non-aromatic AAs, such as β-alanine, glycine, asparagine, glutathione, and threonine, were also found to decrease in HCC tissues (Fig. 3a(ii)). However, arginine increased in the HCC group, which might be attributed to the suppression of the arginine-degrading enzyme arginase I (ARG1), and arginine was also reported to promote tumour growth[41].

The liver is the primary site for the synthesis of lipids and fatty acids. Hepatocyte injury can impair liver function and may result in lipid metabolism dysfunction[42]. For example, except for eicosadienoic acid and nervonic acid, most fatty acids, especially polyunsaturated fatty acids (PUFAs) are down-regulated in HCC tissues (Fig. 3a(iii)). Besides, phosphatidylcholine (PC) is a significant component of cell membranes and may be oxidized by reactive oxygen species[43]. In this study, the levels of PCs tagged with PUFAs were significantly reduced in the HCC group (Fig. 3a(iv)), which might be ascribed to the oxidation of PUFAs in the presence of high oxidative stress in cancer tissue, resulting in a further increase of PCs tagged with saturated fatty acids (SFAs) or monounsaturated fatty acids (MUFAs) (Fig. 3a(v) and Fig. 3d(i)). Furthermore, increases in long-chain acylcarnitines and decreases in short or middle-chain acylcarnitines, such as propionylcarnitine and hexanoylcarnitine, were observed (Fig. 3a(vi)).

Hepatocytes play an essential role were typically involved in nucleotide metabolism. Disorder of some nucleosides, bases and related metabolites was also observed (Fig. 3a(vii–viii)), which were mainly related to purine metabolism. Purine metabolites participating in DNA and RNA syntheses are critical in promoting cell survival and proliferation[44]. Except for inosine, most purine metabolites showed down-regulation in the HCC group, including xanthine, hypoxanthine, xanthosine, deoxyinosine, uridine, uric acid, and adenine (Fig. 3d(ii)), which may be ascribed to decreased activity of related metabolic enzymes[45].

Most saccharides and related metabolites, such as D-ribose, D-sedoheptulose, D-glucuronic acid, D-tagatose, and sucrose, were significantly down-regulated in HCC tissues (Fig. 3a(ix) and d(ii)). However, a high level of the glycolysis metabolite glucose 6-phosphate was observed in the HCC group, while tricarboxylic acid cycle (TCA) metabolites, including fumaric acid and succinic acid, were down-regulated, as previously reported[46]. The changes in these energy-related metabolites suggest rapid glucose expenditure through increased aerobic glycolysis in cancer cells, which could be due to the Warburg effect[47].

In addition, it was observed that several metabolites increased significantly in HCC tissues, including glutathione, 5'-methylthioadenosine, 3,4,5-trimethoxycinnamic acid, oxoadipic acid, and 2-oxoarginine, which have the potential to be biomarkers for liver cancer screening (Fig. 3b and Supplementary Data 2). To compare the above metabolite differences more intuitively, relative changes representation of HCC samples over respective adjacent tissues are shown in Supplementary Fig. 7. Furthermore, the predictive power of the metabolomics was also investigated by the CNN model used for spectral analysis to distinguish HCC tissues from adjacent non-tumour tissues. The accuracy is between 70% and 80%, which is lower than the spectral analysis results, but the accuracy may be improved by increasing the number of samples.

## Raman imaging of tumour margins in liver tissues

Because of the heterogeneity of tumour tissues and differences among patients, variances in Raman data among tissue samples are inevitable (Supplementary Fig. 2), which is challenging for the overall discrimination of different tissues. However, for paired liver cancer and adjacent non-tumour tissues from the same patient, differences in Raman intensity were easily observed in most samples. Therefore, we suggest that the label-free spectrum technology described here can incorporate appropriate image analysis algorithms to visualize cancer margins and facilitate intraoperative tumour delineation.

In order to test this, two liver cancer tissue blocks were selected for Raman scanning. As shown in Fig. 4a and b, the two liver cancer blocks and the corresponding H&E-stained images validated the existence of hepatocyte cancerization in which the hepatic cord had a disordered arrangement with increased cell density and nuclear/cytoplasmic ratio. Bright-field images for the mapping test region of the liver cancer tissues are shown in Fig. 4c. LiveTrack technology was used to continuously adjust the sample height to keep the sample in focus. Surface height data were recorded during the Raman measurements, and three-dimensional (3D) surface profile images of the two tissue samples are shown in Fig. 4c, with maximal height differences of 38.2 and 27.4 μm, respectively. The Raman images were analysed with self-modelling curve resolution (SMCR) and hierarchical cluster analysis (HCA) algorithms. The SMCR method could resolve the unknown Raman mapping dataset into the spectra of pure

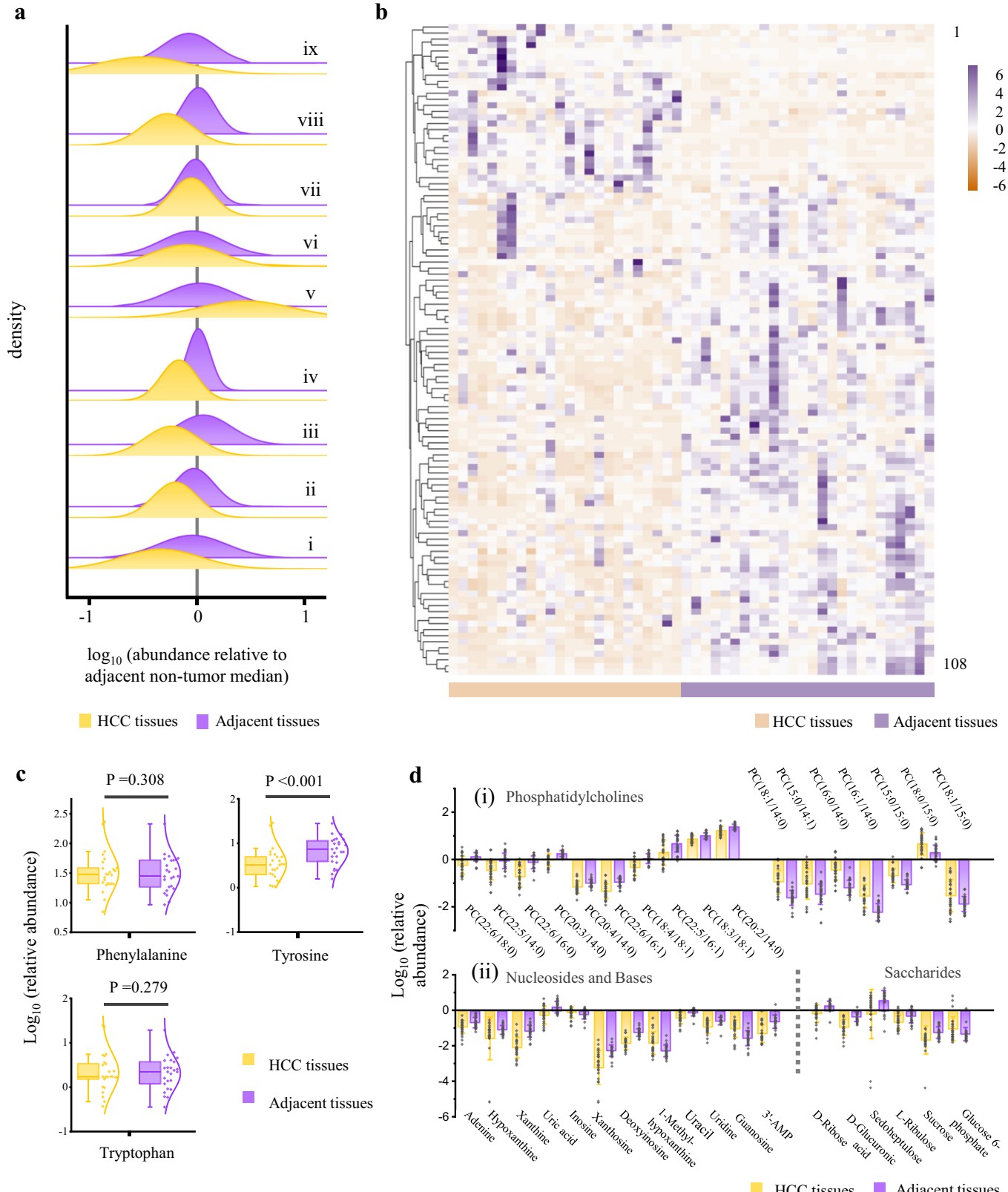

components, yielding concentration images and pure spectrum simultaneously (as described in the "Methods" section). High-quality images were obtained based on the SMCR method (Fig. 4d). A distinct border of the cancerous region could be seen in the first tissue sample, where the carcinoma and hepatic parenchyma regions were successfully distinguished as shown in different pseudocolours. The tumour boundaries were not smooth in the Raman imaging (Fig. 4d and e,

upper panel), probably due to the thin tumour capsule and the small imaging area ($50 \times 50\ \mu m$) with micron-scale scanning interval ($2\ \mu m$). The second tissue showed a relatively poor border of cancer in the displayed area (Fig. 4d and e, bottom panel), intermingling with the liver parenchyma, probably due to the presence of cancerous infiltration, which was barely detectable in brightfield images. These cancerous lesions were confirmed by H&E staining (Fig. 4b). HCA was

**Fig. 3 | Non-targeted metabolomics analysis of HCC tissues and adjacent non-tumour tissues. a** Distributions of relative abundance for nine types of differential metabolites between HCC tissue samples and matched adjacent non-tumour tissue samples, as a ratio to the median relative abundance in the non-tumour tissues. i–ix represent metabolites of tyrosine (i), non-aromatic amino acids (ii), fatty acids (iii), PCs tagged with PUFA (iv), PCs tagged with SFA and MUFA (v), carnitines (vi), nucleosides (vii), bases and their derivatives (viii), and saccharides (ix). **b** Hierarchically clustered heatmap of 108 significantly differential metabolites between HCC tissues and adjacent non-tumour tissues based on Euclidean distance. The blocks were coloured according to the relative expression levels of metabolites. Purple indicates high expression; light orange indicates low expression. **c** Differential content of aromatic amino acids (phenylalanine, tyrosine, and tryptophan) between HCC tissues and adjacent non-tumour tissues. A significant increase in tyrosine was observed in the paracancer group (two-tailed Student's *t*-test, *P* < 0.05), and higher levels of tryptophan and phenylalanine were also observed, but the changes were not significant (two-tailed Student's *t*-test, *P* > 0.05). Boxplots show mean, median, and lower/upper quartiles; whiskers show inner fences. **d** Contents of representative phosphatidylcholines (i), nucleosides, bases, and saccharides (ii) with significant differences between HCC tissues and adjacent tissues. Data are presented as mean values ± SD. HCC tissues, *n* = 25, adjacent tissues, *n* = 25. Source data are provided as a Source Data file.

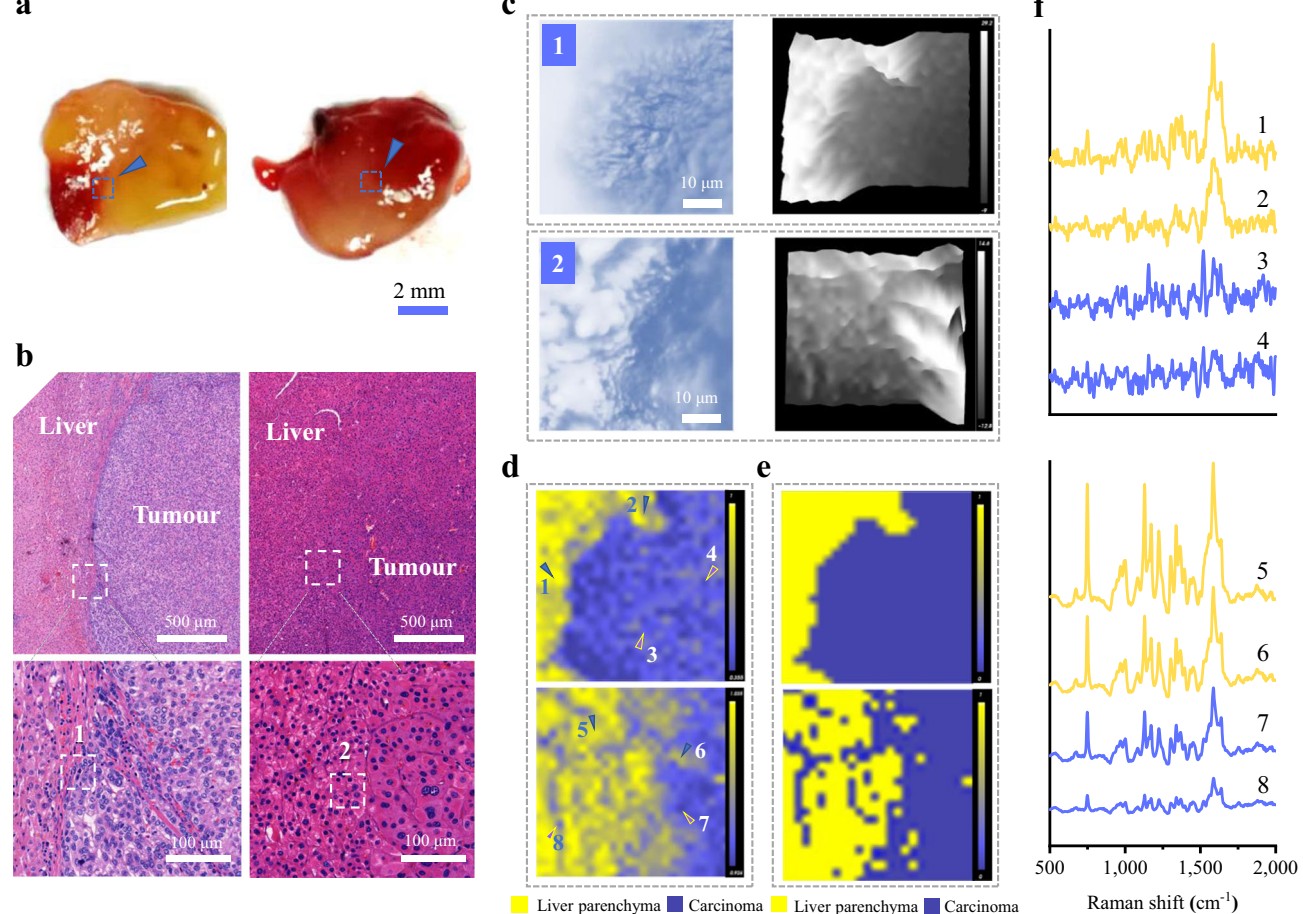

**Fig. 4 | Raman imaging of liver tissue blocks for cancer margin delineation. a** Photographs of two selected liver cancer blocks; arrows point to the positions of the Raman imaging regions. **b** The local H&E-stained images of the two liver tissues and the white boxes in the bottom row are the approximate Raman imaging areas. **c** Bright-field images (50 × 50 μm) of the mapping test region (left) and corresponding 3D surface profile images built with LiveTrack technology (right) of two samples. **d**, **e** SMCR-derived (**d**), and HCA-derived (**e**) Raman images show cancer margins for the liver tissue samples. **f** Typical spectra collected from points 1–8 indicated in (**d**); points 1, 2, 5, and 6 are from the putative paracarcinoma or non-tumour regions and 3, 4, 7, and 8 are from the putative carcinoma regions. Source data are provided as a Source Data file.

another chemometric method used here to combine a set of spectra into clusters with similar spectra in a more abstract manner. The results from HCA-derived images (Fig. 4e) are consistent with those processed with SMCR (Fig. 4d), indicating the reliability of the image-processing algorithms.

To further demonstrate the spectral variation in different liver tissue surface regions, we collected spectra (Fig. 4f) from several locations marked with arrowheads in Fig. 4d. Points 1, 2, 5, and 6 were collected from putative paracarcinoma and non-tumour regions and exhibited higher Raman intensity than those collected from putative carcinoma regions (points 3, 4, 7, and 8) in both tissue blocks. This Raman signature difference provides a basis for image algorithm recognition, supporting our initial expectation of high-precision tumour margin detection using the Raman technique. Furthermore, larger Raman images can also be obtained by using a low-magnification objective and/or increasing the scanning interval with the assistance of the powerful image algorithm (Supplementary Fig. 8).

**Raman imaging of liver tissue slices with different morphologies**
To verify the clinicopathologic and diagnostic capabilities of Raman spectroscopy, we used a micro-Raman spectrometer to image unstained human hepatic tissue slices with a thickness of 5 μm. We first

acquired original spectra in the range of 2000–3400 cm$^{-1}$ (Fig. 5a). The tissue spectra contained a prominent peak at 2930 cm$^{-1}$, which is related to CH$_3$ stretching in proteins[48]. Furthermore, several characteristic Raman peaks related to lipids at 2855, 2885, and 3007 cm$^{-1}$ were detected, which were attributed to symmetric CH$_2$ vibration of lipids, the Fermi resonance or asymmetric vibration of CH$_2$ in saturated straight long acyl chains, and the unsaturated =CH stretch in acyl chains, respectively[49]. Raman image acquisition in StreamHR mode was performed on the unstained liver cancer and adjacent tissue sections at 0.8 μm resolution in the two axial directions. Multivariate analysis was performed on the Raman mapping data to reconstruct spatial distribution maps of the principal chemical components of the tissues. The Raman spectra of pure proteins and lipids were also resolved by the SMCR algorithm (Fig. 5a). SMCR-reconstructed concentration maps of proteins and lipids in normal liver tissue are shown in Fig. 5b and c, respectively. To better understand their relative spatial distribution, a colour overlay image of both is shown in Fig. 5d.

Next, Raman images of typical normal hepatic parenchyma (Fig. 5e) and cancerous areas (Fig. 5f) were compared. The overlay Raman images of proteins and lipids and the corresponding 3D surface profile images, brightfield images, and H&E-stained images of the test area are shown in Fig. 5e and f. The SMCR algorithm-derived images exhibited clear subcellular structure in pseudocolour, revealing variations in lipid and protein concentration. In normal hepatic tissue, proteins were at higher concentrations in the nuclear region, and lipids were mainly distributed in the peripheral regions of the liver cells. While in cancer cells, proteins are mostly distributed near the cell membrane with fewer inside cells. It is worth noting that such differences in the spatial distribution of biochemical components are typically difficult to discern in H&E-stained images. Moreover, typical changes during the transformation of cancer cells, such as the irregular arrangement of hepatocytes and a larger nuclear/cytoplasmic ratio, were also observed in the Raman images, consistent with bright fields and H&E staining. In addition to planar imaging, 3D surface-profile images were obtained, which combined information about chemical composition with the topography of the tissue surface.

In addition to normal and cancerous tissues, Raman scans were carried out on several other tissue regions with typical morphologies, including steatohepatitis, fibrotic, and connective tissues (Fig. 5g–i). These Raman images exhibited various morphological features of cells and tissues, such as fat droplets and filament fibres, which were in accord with the corresponding brightfield and stained images. All individual SMCR-reconstructed concentration maps of proteins and lipids in tissue slices are shown in Supplementary Fig. 9. In addition to two-dimensional (2D) histochemical imaging on the tissue surface, we also generated 'z-stack' images in three dimensions by Raman spectroscopy. Supplementary Fig. 10 shows reconstructed 3D images of the five tissue slices in Fig. 5. The images were reconstructed from a six z-slice stack (5 μm slices) with each plane covering 50 μm × 3 μm, which provided more abundant histochemical depth information of the tissue samples. In this study, the maximum detection depth of liver tissue by confocal Raman spectroscopy under a 532 nm laser was about 200 μm (Supplementary Fig. 11). For deeper tissue detection, the integration of spatially offset Raman spectroscopy (SORS) may be applied to achieve centimetre-level depth detection[50].

### Intraoperative liver cancer diagnosis with a portable Raman system

After validating the performance of Raman spectroscopy in the diagnosis and imaging of liver tissue in vitro, we further investigated its feasibility for real-time intraoperative liver cancer diagnosis. A custom-built handheld portable Raman spectroscopy system was employed intraoperatively to detect hepatic carcinoma. The system was composed of a fibre-coupled laser at 785 nm, a handheld probe, and a fibre spectrometer (details described in the "Methods"

section). The spectrometer was connected to a personal computer with acquisition software, outputting information on the molecular contents of the targeted tissue. During operation, the probe was also covered by a disposable sterile protective cover, which had a negligible effect on the measured Raman spectrum (Supplementary Fig. 12).

Figure 6 shows the average Raman spectra of liver tissue obtained from in vivo intraoperative measurements. The probe was held above the tissue surface to measure Raman signals at several randomly selected points in the carcinoma and adjacent non-tumour regions. The total Raman spectral intensity of the non-tumour region was significantly higher than that of the tumour region. This is consistent with in vitro test results, although there is a difference between tissue spectra collected by the portable Raman spectrometer and micro-Raman spectrometer (Supplementary Fig. 13). The primary spectral differences between tumour and non-tumour regions lie on the protein-related peaks at 640, 976, 1024, 1540, and 1635 cm$^{-1}$, lipids-related peaks at 413, 775, 1314, 1381, and 1436 cm$^{-1}$, and peaks associated with nucleic acids at 689, 1094, and 1514 cm$^{-1}$. As more intraoperative spectral data are acquired, these spectral differences combined with suitable algorithms can help distinguish tumour and hepatic parenchymal regions in surgery, and further intraoperative Raman mapping techniques may make it feasible to visualize tumour boundaries.

## Discussion

Raman spectroscopy has the potential to be a versatile tool for histopathological diagnosis of liver cancer because it allows for rapid detection and high chemical specificity based on intrinsic molecular vibration signals. Specifically, spectra of liver cancer showed weaker overall intensity than those collected from adjacent non-tumour tissues, and different Raman patterns were also observed in varied pathological tissues, which reflected the complexity of biochemical metabolism in liver cancer progression[51]. To confirm these biochemical components' differences between HCC tissues and adjacent tissues, a metabolomics analysis was performed based on LC–MS, revealing that most metabolites exhibited a downward trend in HCC tissues, such as most amino acids, lipids, and nucleic acids, while PCs tagged with SFAs or MUFAs increased. The result was consistent with that of Raman analysis, demonstrating that Raman-based metabolomics, also known as Ramanomics[52], could bring comprehensive and reliable biological information as traditional metabolomics, and distinguish different pathological tissues more conveniently and cost-effectively without additional consumables.

In addition, a VGG-16-based CNN model was built and successfully employed in the distinction between Raman spectra collected from hepatic carcinoma tissues and adjacent non-tumour tissues and the recognition of different hepatic pathological tissues, including different subtypes, tumour stages, and differentiations. The results demonstrated that Raman spectroscopy combined with deep learning can accurately record and identify spectral patterns in different pathological samples. Furthermore, we also plan to study the discrimination of HCC precursors and non-malignant liver lesions in the follow-up work, as well as the distinction between primary and secondary liver cancer, which is crucial to the treatment and prognosis of hepatic carcinoma.

Based on the Raman spectral differences, the cell morphology of tissue slices can be depicted in a label-free way. The SMCR-resolved Raman images could not only show spatial distributions but also quantitative identification of the principal biochemical components (proteins and lipids) of cells and tissues at 2D and 3D subcellular scales, which are not applicable by standard H&E-staining methods. This indicates that the subcellular Raman analysis has great potential for simplifying cancer diagnosis during clinical trials and provides a perspective on histopathological diagnosis.

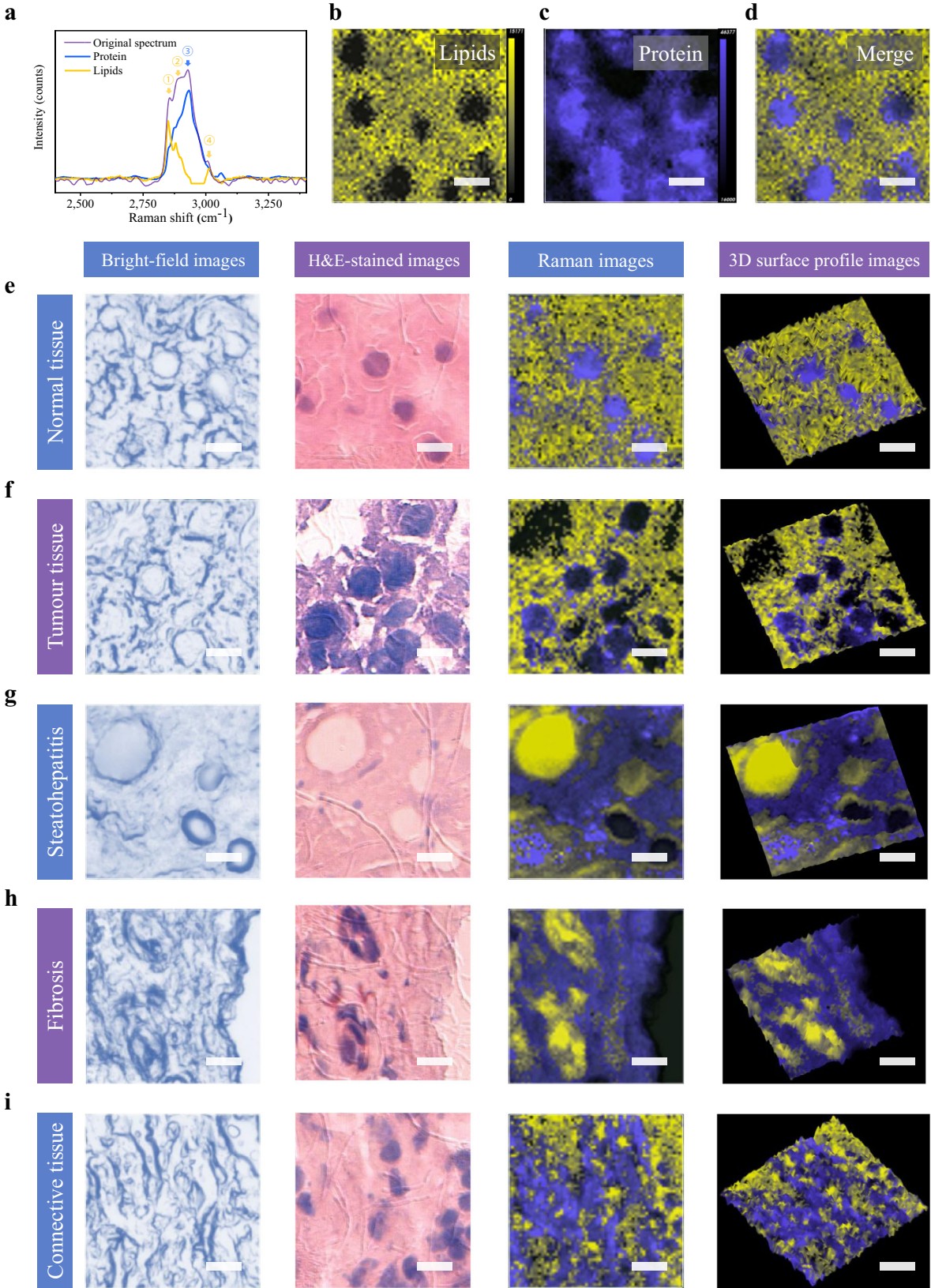

**Fig. 5 | Raman imaging of various liver tissue slices. a** Typical Raman spectrum of liver tissue slices in the range of 2000–3400 cm⁻¹ (purple line); the Raman spectra of pure protein (blue line) and lipids (yellow line) are also resolved by the SMCR algorithm. The Raman peak at 2930 cm⁻¹ (arrow 3) is related to protein, and peaks around 2855 cm⁻¹ (arrow 1), 2885 cm⁻¹ (arrow 2), and 3007 cm⁻¹ (arrow 4) are characteristic of lipids. **b**–**d** SMCR-reconstructed concentration maps of lipids (**b**) and protein (**c**) in normal liver tissue; **d** is an overlay of the two. To merge the two images, the minimum LUT value of yellow (lipids) was adjusted. **e**–**i** Bright-field images of the mapping test area and corresponding H&E-stained, SMCR-derived Raman, and corresponding 3D surface profile images of normal liver tissue (**e**). **f**–**i** show other tissue regions with typical morphologies, including cancerization (**f**), steatohepatitis (**g**), fibrosis (**h**), and connective (**i**) tissues. All scale bars are 10 μm.

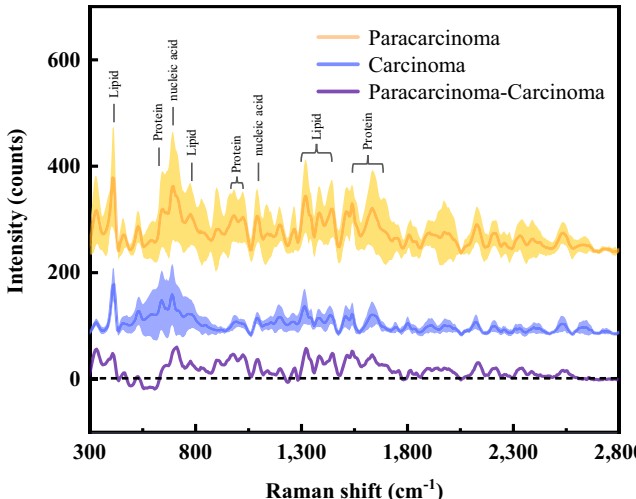

**Fig. 6 | Intraoperative detection with a handheld fibre-coupled Raman spectrometer.** The average Raman spectra of in vivo intraoperative measurements for carcinoma and paracarcinoma tissue were collected from six patients. The spectra were collected by a handheld portable Raman spectrometer system equipped with a 785 nm NIR laser and a computerized CCD spectrometer. The shaded areas represent standard deviations of the means. Source data are provided as a Source Data file.

Clinically, surgical resection is a standard method for cancer treatment. Accurate tumour boundary identification is helpful for the complete resection of lesions without excessive resection of normal tissue, especially for hepatic metastases resection. However, this is often a challenge for surgeons without suitable intraoperative methods for visually distinguishing the two tissues. We employed Raman spectroscopy in both ex vivo and in vivo measurements for liver tissues characterized by different pathological patterns. Images acquired with distinct cancer margins were visible based on the spectra differences between tumour and hepatic parenchymal regions. Besides, we successfully verified the feasibility of the portable Raman spectrometer to distinguish tumours from non-tumour regions during surgery. These suggest that the Raman technique has the potential to assist surgeons in rapidly analysing regions of interest during surgery, without disruptions or delays from intraoperative frozen sectioning or H&E staining[53].

It is noteworthy that Raman spectra of tissue collected by the portable Raman spectrometer are at variance with those obtained from the micro-Raman spectrometer, specifically at some peak positions (Supplementary Fig. 13). This may be attributed to differences between the two types of spectral equipment, such as laser sources, laser power and wavelength, and spectrometer. A similar result was also observed in a Raman detection of mouse tumours by a portable Raman device[54]. However, although there were differences in the Raman data measured by the two pieces of equipment, this did not affect Raman spectroscopy's ability to discriminate cancerous from adjacent normal tissues.

Moreover, the practical application of Raman spectroscopy as a clinical tool still requires further exploration and optimization. In this research, the Raman signal derived from liver tissue blocks is high enough to be detected for diagnosis. Thus, no metallic nanoparticles were required for Raman signal enhancement, avoiding the risk of metal particle toxicity and excretion in clinical applications[55,56]. However, one of the primary defects of spontaneous Raman spectroscopy is the relatively low signal intensity, necessitating a compromise between image quality and short acquisition time. This might be solved by combining the coherent Raman spectroscopy, which is based on non-linear optical effects and can be used simultaneously for

high speed and high spatial resolution in Raman spectral imaging[48,57]. In addition, the limited spectral collection points with the handheld Raman probe may lead to missing lesions during surgery, while the respiratory fluctuations may affect the spectral quality even though the Raman integration time is more than 10-fold shorter than the respiratory period. We hope to demonstrate that an intelligent robot collaboration system can be used to assist in intraoperative Raman imaging to solve such issues in our follow-up work. Furthermore, the development of integrated handheld diagnostic equipment has allowed for the precise removal of regions of interest in a more convenient manner during cancer surgery[58]. However, with the emergence of these new instruments, standardization of instruments across all users is also of concern[29], and factors such as excitation wavelength, laser power, spectrum collector type, along with data processing algorithms, should be standardized. Therefore, it is hoped that a convenient Raman spectroscopy system with a normalization criterion for the accurate diagnosis will be developed soon to facilitate its clinical adoption.

Although the current work was carried out in the context of hepatic carcinoma, the same approach could be used to assess similar histologic features of tumours in other organs. Thus, we conclude that the Raman technique, coupled with intelligent algorithms, could be applied for the diagnosis of liver and other types of tumours, playing a potential role in pathological identification and intraoperative guidance.

## Methods

### Ethics statement

All current research methods were carried out under guidelines approved by the Ethics Committee of The First Affiliated Hospital of Wenzhou Medical University (Ethical Approval No. 2020213). Informed written consent was obtained from all patients for whom the risks and benefits of the trial were detailed.

### Preparation of human tissue samples

In the present study, a total of 240 tissue samples with paired liver cancer and adjacent non-tumour tissue blocks were obtained from 120 patients with primary liver cancer, of whom 98 individuals were diagnosed with hepatocellular carcinoma (HCC), and 22 were intrahepatic cholangiocarcinoma (ICC). Detailed information about the clinical and histopathological diagnoses of the 120 patients is provided in Supplementary Table 1. Diagnostic results of patients, including cancer type, cancer stage, and differentiation type were confirmed based on related clinical and pathological indicators[4] by doctors from The First Affiliated Hospital of Wenzhou Medical University.

After surgical resection and inspection, all samples were stored in a refrigerator at −80 °C. Before spectral measurements, tissue blocks were placed on a glass slide, and water from the tissue surface was absorbed with tissue paper. Minimal sample processing was applied so as to facilitate further application in intraoperative detection and imaging. The tissue blocks collected from cancer tissues and adjacent non-tumour tissues were reconfirmed by pathologists based on H&E staining.

For the imaging of tissue slices, the tissue slices of thickness 5 μm were prepared using a freezing microtome and the slices were attached to glass slides for Raman measurements. The same tissue was observed by H&E staining after Raman measurements.

### Raman measurements and imaging of liver tissue

Raman spectra of tissue samples were obtained by a micro-Raman spectrometer (Renishaw, Gloucestershire, UK) using an excitation laser at 532 nm. The laser beam was focused onto the sample's surface by an L×50 objective (numerical aperture (NA) = 0.50, the working distance (WD) = 8.2 mm). Each spectrum was integrated over 3 s at 5% (1.25 mW cm$^{-2}$) laser power. At least 50 spectra in the range of

500–2000 cm$^{-1}$ were collected from randomly selected points on the surface of each tissue sample. Before statistical analysis, the spectral data were processed using WIRE 5.3 software with baseline subtraction and Savitzky–Golay smoothing to remove the fluorescence background and increase the signal-to-noise ratio.

In Raman imaging of tissue blocks for tumour margin delineation, Raman spectra were also acquired with an L×50 objective (NA = 0.50, WD = 8.2 mm), equipped with a 532 nm laser, with 2.5 mW cm$^{-2}$ laser power and 2 s exposure time for each data point. The Raman scans were collected with a resolution of 2 µm in the x- and y-directions (StreamHR mode), which allows for rapid collection of spectra at high spatial resolution. To avoid compromising the quality of high magnification Raman analysis because of uneven surfaces of tissue samples during imaging, LiveTrack focus-tracking technology was used to automatically focus on the sample surface during image acquisition.

In Raman imaging of tissue slices, Raman spectra were acquired at a higher resolution of 0.8 µm in both axial directions, with 12.5 mW cm$^{-2}$ laser power and 0.5 s exposure time for each data point. StreamHR and LiveTrack mode equipped with a 532 nm laser and L×50 objective (NA = 0.50, WD = 8.2 mm) were used as above for rapid and precise data acquisition.

WiRE 5.3 software self-modelling curve resolution (SMCR) and hierarchical cluster analysis (HCA) were used for Raman imaging analysis of tumour margins. The spectra were pre-processed by baseline correction, smoothing, cosmic ray removal, and noise filtering before any further multivariate imaging.

### Intraoperative Raman detection of liver cancer

A handheld Raman spectroscopy system was employed in intraoperative liver cancer detection. A fibre-coupled laser at 785 nm (FC-D-785, Changchun New Industries Optoelectronics Technology Co., Ltd., China) was employed as the laser source, which was introduced into a handheld probe connected to a 100 µm fibre for laser excitation and a 200 µm standard fibre for signal collection (NA = 0.22). Raman signals were collected with a charge-coupled device (CCD)-based fibre spectrometer (QE Pro, Ocean Optics Inc., Dunedin, FL, USA) over a spectral range of 200–1100 nm and resolution of 6–7 cm$^{-1}$. The spectra covered a wide range of spectral shifts from 0 to 4000 cm$^{-1}$. The spectrometer was connected to a PC by an OceanView interface.

To avoid intraoperative infection, the Raman probe and connected fibre were wiped with medical alcohol and covered with a disposable sterile protective slipcover made of polyethylene (Renhe Medical Supplies Industry and Trade Co., Ltd, Chun'an, China). Before measurement, the tissue surface was processed to minimize blood in the tested area. A background spectrum was first recorded and subtracted automatically before every measurement using an integration time of 0.2 s with the laser off. Then five measurements were taken with an integration time of 0.2 s from randomly selected points of each tissue surface. The laser power at the tip of the probe was about 40–56 mW cm$^{-2}$, measured with an optical power meter (PM100D from Thorlabs Inc.). The measured tissues were collected for tissue biopsy by pathologists after surgery.

### Tissue metabolomics analysis by LC–MS

Twenty-five pairs of matched HCC tissues and adjacent non-tumour tissues were selected to detect metabolic differences by LC–MS in this study. The specific sample processing and testing methods were as follows.

**Extraction of metabolites.** 25 mg of sample was weighed in an Eppendorf tube, and 500 µl of extract solution (methanol:acetonitrile:water = 2:2:1) and isotopically labelled internal standard mixture were added. Then the samples were homogenized at 35 Hz for 4 min and sonicated for 5 min in an ice-water bath. The homogenization and sonication cycle

was repeated three times. Then the samples were incubated at −40 °C for 1 h and then centrifuged at 12,000 rpm for 15 min at 4 °C. The resulting supernatant was transferred to a fresh glass vial for analysis. The quality control (QC) sample was prepared by mixing equal aliquots of the supernatants from all the samples.

**LC–MS/MS analysis.** LC–MS/MS analyses were performed using a UHPLC system (Vanquish, Thermo Fisher Scientific) with a UPLC BEH Amide column (2.1 mm × 100 mm, 1.7 µm) coupled to a Q Exactive HFX mass spectrometer (Orbitrap MS, Thermo). The mobile phase consisted of 25 mmol/L ammonium acetate and 25 mmol/L ammonia hydroxide in water (pH = 9.75) (A) and acetonitrile (B). Gradient elution was used: 0-0.5 min, 95% B; 0.5–7 min, 95%–65% B; 7–8 min, 65%–40% B; 8–9 min, 40% B; 9–9.1 min, 40%–95% B; 9.1–12 min, 95% B. The flow rate was 0.5 ml/min. The auto-sampler temperature was 4 °C, and the injection volume was 2 µl.

The QE HFX mass spectrometer could acquire MS/MS spectra in information-dependent acquisition mode in the control of the acquisition software (Xcalibur, Thermo). In this mode, the acquisition software continuously evaluates the MS spectrum. The ESI source conditions were as follows: sheath gas flow rate, 30 Arb; Aux gas flow rate, 25 Arb; capillary temperature, 350 °C; full MS resolution, 60,000; MS/MS resolution, 7500; collision energy, 10/30/60 in NCE mode; spray voltage, +3.6 or −3.2 kV. The measurement was achieved with the support of Shanghai Biotree Biotech Co., Ltd.

**Data preprocessing and annotation.** Raw data were converted to mzXML format using ProteoWizard and processed by an in-house programme developed using R and based on XCMS for peak detection, extraction, alignment, and integration. The identification of compounds was based on the mass-to-charge ratio of the parent ions in the primary mass spectrometry and characteristic product ions generated by the fragmentation. A commercial MS2 database (BiotreeDB) was used for annotating metabolites. The cutoff for annotation was set at 0.6. All the detected ions were normalized according to internal standards for subsequent quantitative analysis. Ions detected from ESI$^+$ and ESI$^-$ were imported into SIMCA software (Umetrics, Umea, Sweden) for multivariate analysis. To screen differential metabolites between HCC and adjacent non-tumour tissues, ions with significant changes (Student's t-test, $P < 0.05$), as well as variable importance in the project (VIP) values > 1 were selected in the OPLS-DA model. Ultimately, 57 metabolites in ESI$^+$ mode and 51 in ESI$^-$ mode were identified and selected as candidate differential metabolites.

**CNN architecture and training details.** The Raman spectra were analysed by PyTorch, and the CNN architecture was modified based on the VGG-16 framework[36]. The CNN model was built up with 13 one-dimensional convolutional layers (including 2 convolution layers with 64 kernels, 2 convolution layers with 128 kernels, 3 convolution layers with 256 kernels, and 2 sets of 3 convolution layers with 512 kernels), 5 pooling layers (size 2, stride 2), and 3 fully connected layers, as illustrated in Fig. 2h.

To differentiate spectra from cancer and paracancer tissue areas, a soft-max optimizer was used to transform the output of the previous connection layer into a probability output, with a learning rate of 0.0001, and a batch size of 128. To prevent overfitting, a dropout layer was employed with a rate of 50%, which nullified the contribution of 50% of neurons towards the next layer to reduce over-reliance on certain neurons for classification. Spectral data from 20 pairs of liver tissue samples were randomly selected as a test set, and spectral data of the remaining 100 pairs of samples were randomly divided into a training set and a validation set in a ratio of 8:2. Similar splits were performed in other three classification models that 20% of the samples in each group are randomly selected as the

test set, and the remaining samples are divided into the training set and the validation set according to the ratio of 9:1. Weight balancing was used to alleviate possible data imbalances by altering the weight of each training sample when computing the loss to make all classes contribute equally to the loss (detail of the code can be found in the following GitHub link).

Correlation analyses of representative Raman peak intensities based on Pearson's correlation coefficient ($r$) in the form of a correlation matrix heatmap were calculated by Python. The version of software used for analysing data are as follows: Python 3.9.9, pytorch 1.10.1, numpy 1.22.0, matplotlib 3.5.1, torchvision 0.11.2, tqdm 4.62.3, pandas 1.0.4, and seaborn 0.9.0.

### Statistics and reproducibility
Data are expressed as mean ± standard deviation (SD) as indicated in each figure legend. All data conformed to the normal distribution. The statistical significance between the two groups was obtained by a two-tailed Student's $t$-test with $P$ values < 0.05 considered significant. The micrographs of staining images and Raman images are representative of three independent measurements.

### Images processing
The SMCR algorithm was used in tissue slice Raman imaging analysis to identify various hepatic histopathological patterns. The SMCR method is a form of multivariate curve determination and alternating least-squares analysis, which transforms component information into physically meaningful components. Briefly, SMCR decomposes the experimental data matrix (**X**), containing all the spectral data for every pixel, into two smaller matrices, the matrix of concentration imaging (**C**) and the matrix of the pure spectrum (**S**):

$$\mathbf{X} = \mathbf{C} \cdot \mathbf{S^T} + \mathbf{E} \tag{1}$$

where **E** is the error matrix. By initially estimating the spectra matrix **S**, **C** and **S^T** can be calculated, and then iterative optimization using the alternative least-squares (ALS) algorithm can be performed until convergence is reached[59].

### Reporting summary
Further information on research design is available in the Nature Portfolio Reporting Summary linked to this article.

## Data availability
The data supporting the results of this study are available within the manuscript and its Supplementary Information. The raw mass spectrometry data associated with this manuscript can be found in Supplementary Data 1 and 2. While the commercial MS2 database (BiotreeDB) needs to be obtained by contacting Biotree Biotech Co., Ltd. Source data are provided with this paper.

## Code availability
The code used for the CNN model and demo datasets is available at https://github.com/thidoiSanren/CNN_liver-cancer_Raman[60].

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

## Acknowledgements

This work was financially supported by Zhejiang Provincial Natural Science Foundation of China (Grant No. LR19H180001) to Y.W., Leading Talent Innovation and Entrepreneurship Project of Wenzhou (RX2016005) to Y.W., and Public Projects of Wenzhou (2020005) to Y.W.

## Author contributions

L.H., Q.Z. and Y.W. designed and conducted the study. H.S. and K.S. provided clinical samples and assisted in tissue staining and intraoperative operating. L.S., Y.C. and Y.G. performed statistical analyses and established the CNN models. X.R. and D.J. helped L.P. with tissue preparation and the Raman test. L.P. was responsible for data processing and graphing. L.H, X.L. and Q.Z. wrote the manuscript. W.K., Q.Z. and Y.W. supervised the project. All authors reviewed the manuscript.

## Competing interests

The authors declare no competing interests.
