## [Peer Review File · Nature Communications]

REVIEWER COMMENTS

Reviewer #1 (Remarks to the Author): expert in Raman spectroscopy

The paper "Rapid, label-free histopathological diagnosis of liver cancer based on Raman spectroscopy and deep learning" by Huang presents a strong technical study and demonstration for in vitro and intraoperative pathological diagnosis of liver cancer by Raman spectroscopy technology. The methods are clearly presented and validated, the combination of Raman spectroscopy with deep learning is exciting, and the demonstration of the portable handheld Raman system is impressive. The implication of this work to pathology and medical practice could be large.

However, there are a few important technical issues that need clarification and re-examination.

1. For in vitro studies, the area of the liver tissues is over mm^2 . This is ordered of magnitude larger than what spontaneous Raman imaging can cover. How does this mismatch affect the result?
2. For intraoperative diagnosis with the portable Raman system, again one can only acquire a limited number of Raman spectra on pre-selected points instead of mapping the entire region of interest. With such under-sampling, would one worry about that one might miss the actual lesion during the operation?

Reviewer #2 (Remarks to the Author): expertise in liver cancer

This is a translational research to develop and validate a workflow for in vitro and intraoperative pathological diagnosis of liver cancer by utilizing Raman spectroscopy to human hepatic tissue samples. The authors established the method to distinguish carcinoma tissue from non-tumor tissue by using Raman spectroscopy combined with deep learning. They also developed a portable handheld Raman system and applied for real-time intraoperative human liver cancer diagnosis. However, the following concerns need to be addressed from a clinical point of view:

Major and minor comments

1. On Page 7, the authors judged tumor stage and differentiation grade by using Raman spectroscopy and described the judgement is crucial for providing effective and appropriate cancer treatment. However, the size and the number of tumors and the presence of vascular infiltration determine the stage of liver cancer. Is it possible to determine these by Raman spectroscopy? In addition, tumor differentiation usually does not affect treatment strategy.
2. On Page 10, the authors described tumor heterogeneity posed greater challenges to discrimination of different stages and differentiation grades of tumor tissues, with accuracies of 79.2% and 75.7% and to judgement of the liver cancer subtypes HCC and ICC, yielding an identification accuracy of 86.3%. Are these results of diagnostic accuracy better than traditional CT, MRI and US?
3. On Page 10, the authors mentioned about the sensitivity of serological AFP. It is well known that AFP alone is not enough to diagnose with HCC, which is usually diagnosed by US, CT or MRI. Is Raman spectroscopy superior to these modalities?
4. In Figure 4, the authors showed cancer margin delineation. In HCC, the tumor capsule clearly distinguishes between cancerous and non-cancerous tissues. Boundaries were not smooth in Raman images like Fig. 4d and 4e. Can't you see the tumor capsule in Raman images?
5. In Figure 6, the authors showed a preliminary result of usage of a handheld fibre-coupled Raman spectrometer. Do we need Raman spectrometer to observe tumors that can be seen by inspection or intraoperative ultrasound? Isn't it difficult to observe at the micro level because of camera shake and respiratory fluctuations?
6. Liver cancer often occurs deep in the liver. How deep can you observe in Raman spectroscopy?
7. On Page 22, the authors described that Ramanomics could distinguish different pathological tissues more cost-effectively. How much does it cost to observe in Raman spectroscopy?

8. In Figure 2, legends of 2c and 2d are opposite. Fig. 2c is for tumor stage and Fig. 2d for tumor differentiation.
9. "Tumor" and "tumour" are mixed in the article.

Reviewer #3 (Remarks to the Author): expert in deep learning network analysis

This paper proposed a new system for histopathological diagnosis of liver cancer based on 2 Raman spectroscopy and deep learning. First, a CNN-based DL pipeline is used to classify the spectral data into paracancer or cancer tissues. The model is further used to differentiate different pathological types of liver cancer tissues. Last but not least, a portable Raman system was employed in surgery to validate this pipeline.

While the whole work is very novel and potentially impactful to bring benefits in liver cancer diagnosis, the study suffers from the following limitations:

- 1) The spectral data analysis. A liver tissue Raman database was established with 50 spectra per tissue sample, and a total of 12,000 spectra were obtained from 120 pairs of liver tissue samples. When training the VGG-model, the dataset is randomly splitted into 80% training, 10% validation and 10% test. However, due to the potential correlations of raman spectral data at the same tissue, a purely random split of training and testing data can cause label leakage, and therefore inflate the accuracy. A proper split would be following: take a random 100 pairs of liver tissue samples and all the raman spectral data on these tissue samples as training, and use the spectral data on the remaining 20 pairs of liver tissue sample as testing. This design will ensure the fair evaluation of DL-based models without potential label leakage. Further, the compared method, OPLS-DA is quite a weak comparison in my view. It would really demonstrate the power of DL-based approach by comparing with stronger baselines such as random forest, xgboost.
- 2) The subsequent experiments of CNNs are conducted to classify multiple clinical phenotypes such as tumor stages, HCC/ICC subtypes. Similar to the concerns of potential label leakage mentioned above, the data splits need to be re-considered to justify the results. Further, to better understand the method, details about how to preprocessing the images, and how to augment data, how to alleviate the imbalance of labels should be discussed.
- 3) Tissue metabolomics analysis. The authors find a few metabolic features that can distinguish HCC and adjacent tissues. From Figure 3a, 3b, there are generally two types of metabolomics signatures: one is enriched in HCC and the other enriched in adjacent tissues. While the pattern is clear, the predictive power of these metabolomics is not explored. Further, it would be very interesting and informative to compare the predictive power of these metabolomic features with the deep learning models developed using the spectral data.
- 4) The intraoperative liver cancer diagnosis using a portable Raman system is extremely impressive! However, while the authors show different patterns of spectral Raman shift for different cancer tissues, the connection with previously developed DL models is missing. It is unclear whether the authors had validated the trained DL models using this portable Raman system, which would be extremely valuable to do so as a true validation of generalization.
- 5) Transparency of reproducibility. The github link provided in the manuscript simply contains the codebase of plain VGG model training scripts. Without the spectral/imaging data, it is impossible for other researchers to validate the code or deploy the code. While I understand the privacy concerns of releasing patient data, I would highly encourage the authors to at least provide de-identified or simulated data so that other researchers can test/validate the proposed methods.

Reviewer #4 (Remarks to the Author): expert in metabolomics

The manuscript by Huang et al, describes a methodology to use Raman spectroscopy and deep learning to identify liver cancer tissue. The goal is to apply this in the surgery theatre for intraoperative pathological diagnosis (compared to classical biopsy methods) to better resect cancer tissue.

The analyses were also compared to metabolomics analyses.

Focusing mainly on the Metabolomics analysis, the performed overall analysis looks robust. I have two comments:

1. The data representation in e.g. panel 3d appears not very intuitive. I am wondering if a fold change representation of HCC sample over respective adjacent tissue would not be more intuitive (in such case, the signal for each metabolite from adjacent tissue would always be = 1).
2. The analytical method is optimally suited for polar metabolites while lipids etc are barely retained by the used chromatographic method (all hydrophobic compounds elute at very early retention time). I am wondering if these compounds can be detected with confidence when barely separated? What are the measures applied to allow compound identification?

REPONSE TO REVIEWERS' COMMENTS

Reviewer #1 (Remarks to the Author): expert in Raman spectroscopy

The paper “Rapid, label-free histopathological diagnosis of liver cancer based on Raman spectroscopy and deep learning” by Huang presents a strong technical study and demonstration for in vitro and intraoperative pathological diagnosis of liver cancer by Raman spectroscopy technology. The methods are clearly presented and validated, the combination of Raman spectroscopy with deep learning is exciting, and the demonstration of the portable handheld Raman system is impressive. The implication of this work to pathology and medical practice could be large.

However, there are a few important technical issues that need clarification and re-examination.

1. For in vitro studies, the area of the liver tissues is over mm^2 . This is ordered of magnitude larger than what spontaneous Raman imaging can cover. How does this mismatch affect the result?

[Response]:

Thank you for your comment. As shown in Figure 4, the area selected for Raman imaging is $50 \times 50 \mu\text{m}$, the Raman scanning step size is $2 \mu\text{m}$ in x- and y-directions and the integration time of each acquisition is 2 s. In general, a large Raman imaging area usually means a longer scanning time. However, under the premise of ensuring the required imaging resolution and Raman signal-to-noise ratio, we can increase the scanning step size or reduce the integration time to achieve large-area imaging and shorten the imaging time.

For example, in several recent studies, Raman imaging of tumours in vivo or in vitro in mice was performed with low-magnification objectives or fibre-optic probes at millimetre or submillimetre scan intervals (*ACS Nano* 2018, 12, 9669-9679 and *ACS Nano* 2017, 11, 1488-1497). Moreover, for liver tumour resection, the imaging resolution of the tumour boundary at the millimetre level can meet the clinical requirements.

In order to demonstrate that a larger area of Raman imaging can be obtained, we used a 5× objective lens with a scanning interval of 50-100 μm to acquire Raman signals on the surface of the tumour block and obtained a Raman image of about 1.5 mm² to delineate the tumour boundary with the assistance of the powerful image algorithm. Furthermore, the imaging of a larger area can be further obtained by imaging splicing or by using fibre optic probes.

Figure R1-1. A liver cancer tissue block (a), and its bright-field image (b) and corresponding Raman image (c) revealing the cancer boundary. Raman spectra were acquired with a 5× objective (NA = 0.12, WD = 14 mm), equipped with a 532 nm laser, with 2.5 mW cm⁻² laser power and 2 s exposure time for each data point. The Raman scans were collected with a resolution of 50 μm and 100 μm in the x- and y-directions (horizontal axis and vertical axis), respectively.

In addition, the development of emerging Raman technology is also expected to solve the above problems. For example, non-linear coherent Raman spectroscopy can be used for fast Raman imaging, and a large area of Raman imaging can be obtained in a short time. However, this also puts forward requirements of advanced/expensive spectroscopic equipment, which is not within the scope of the current study.

[Action taken]:

Supplementary Figure 8 (shown above as Figure R1-1) has been added to Supplementary information as follows.

Supplementary Figure 8. A liver cancer tissue block (a), and its bright-field image (b) and corresponding Raman image (c) revealing the cancer boundary. Raman spectra were acquired with a 5× objective (NA = 0.12, WD = 14 mm), equipped with a 532 nm laser, with 2.5 mW cm⁻² laser power and 2 s exposure time for each data point. The Raman scans were collected with a resolution of 50 μm and 100 μm in the x- and y-directions (horizontal axis and vertical axis), respectively.

The sentence has been added in the last paragraph on page 16, “Furthermore, larger Raman images can also be obtained by using a low-magnification objective and/or increasing the scanning interval with the assistance of the powerful image algorithm (Supplementary Fig. 8).”

2. For intraoperative diagnosis with the portable Raman system, again one can only acquire a limited number of Raman spectra on pre-selected points instead of mapping the entire region of interest. With such under-sampling, would one worry about that one might miss the actual lesion during the operation?

[Response]:

Thank you for pointing out this concern. This work provides proof-of-concept applications with the Raman system for intraoperative tumour diagnosis. However, for the large area of mapping, indeed one would rely on continuous point scanning. Therefore, at the current status, surgeons could employ imaging techniques such as intraoperative ultrasound or rely on their own experience to prejudge the lesion area. Meanwhile, the Raman spectra can be used as assistance for the diagnosis of suspected

lesions to obtain more accurate information about tumour boundaries or lesion regions to avoid the miss of lesions.

Besides, in another follow-up work, we are building an automatic robot system handling the optical fibre probe to automatically scan the patient tissue for intraoperative Raman mapping. The robot gripper holding the optical fibre probe can move point by point on the plane according to previously set commands. The system is equipped with a laser distance sensor, which helps to adjust the probe height according to the surface roughness thus fixing the distance between the probe and tissue surface during scanning to obtain the Raman spectra of the tissue surface. We expect the follow-up work could carry out Raman mapping during surgery, and help doctors achieve more accurate lesion identification.

[Action taken]:

The following sentences have been added on page 25, “Besides, the limited spectral collection points with handheld Raman probe may lead to the miss of lesions during surgery, while the respiratory fluctuations may affect the spectral quality even though the Raman integration time is more than 10-fold shorter than the respiratory period. We expect that an intelligent robot collaboration system can be used to assist in the intraoperative Raman imaging to solve such issues in our follow-up work.”

Reviewer #2 (Remarks to the Author): expertise in liver cancer

This is a translational research to develop and validate a workflow for in vitro and intraoperative pathological diagnosis of liver cancer by utilizing Raman spectroscopy to human hepatic tissue samples. The authors established the method to distinguish carcinoma tissue from non-tumor tissue by using Raman spectroscopy combined with deep learning. They also developed a portable handheld Raman system and applied for real-time intraoperative human liver cancer diagnosis. However, the following concerns need to be addressed from a clinical point of view:
Major and minor comments

1. On Page 7, the authors judged tumor stage and differentiation grade by using Raman spectroscopy and described the judgement is crucial for providing effective and appropriate cancer treatment. However, the size and the number of tumors and the presence of vascular infiltration determine the stage of liver cancer. Is it possible to determine these by Raman spectroscopy? In addition, tumor differentiation usually does not affect treatment strategy.

[Response]:

Thank you for your comment. Clinical staging of HCC is mainly diagnosed based on imaging characteristics, including the number and size of HCC nodules, the presence of vascular invasion and extrahepatic metastases. In this study, Raman spectra collected from tissue samples can reveal the alterations of the biochemical molecules in the tissue during carcinogenesis, which could be various for different tumour stages and differentiation types. However, it is not easy to directly show the size and number of nodules at the macro level only through spectral information.

Vascular invasion includes macrovascular invasion (MaVI) and microvascular invasion (MVI), where microvascular invasion is usually identified based on microscopy. As cancer progresses, the biochemical composition of cancer tissue changes accordingly. There are studies demonstrating that serum and tissue biomarkers, such as α -fetoprotein (AFP) and des- γ -carboxy-prothrombin (DCP), could be used to

predict MVI (*Liver Int.* 2014, 34, 937-941 and *J. Cancer. Res. Clin. Oncol.* 2017, 143, 293-303). The Raman signal of these biomarkers may be collected from tissue samples to enable the judgement of MVI based on spectral analysis.

Therefore, we supplemented the data analysis of Raman spectroscopy for MVI judgment. In all 120 patients with liver cancer, we obtained the MVI status of 84 patients, of which 61 were M0 (without MVI), and the remaining 23 were M1 or M2. We tried to judge the presence of MVI in patients by Raman spectra of tissues. The accuracy to differentiate M0 and M1/M2 is 66.9% based on the deep learning model and the AUC value is 0.694 (Figure R2-1). Given that Raman spectra were collected from local tissue blocks, further broadening the sources of the spectral collection might improve the ability of Raman spectroscopy to determine the presence of vascular infiltration in liver tissue.

In this study, we expected to explore the relationship between the biochemical composition of tissue samples and cancer stage by Raman spectra. The results show that Raman spectroscopy has feasibility for the discrimination of HCC stage and differentiation grade.

Figure R2-1. The ROC curve of the classification of the grade of microvascular invasion (MVI) based on Raman spectra and the VGG-16 model. In all 120 patients with liver cancer, the MVI statuses of 84 patients were indicated, of which 61 were M0 (without MVI), and the remaining 23 were M1 or M2. The accuracy to differentiate M0 and M1/M2 is 66.9% based on the deep learning model and the ROC value is 0.694.

Tumour differentiation grade is one of the factors for evaluating the recurrence and prognosis of liver tumours. Although differentiation grade results may not directly affect treatment strategy, to better evaluate the ability of Raman spectroscopy to reveal pathological conditions, we also include tumour differentiation grades into the scope of the study.

[Action taken]:

Supplementary Figure 6 (shown above as Figure R2-1) has been added to Supplementary information as follows.

Supplementary Figure 6. The ROC curve of the classification of the grade of microvascular invasion (MVI) based on Raman spectra and the VGG-16 model. In all 120 patients with liver cancer, the MVI statuses of 84 patients were indicated, of which 61 were M0 (without MVI), and the remaining 23 were M1 or M2. The accuracy to differentiate M0 and M1/M2 is 66.9% based on the deep learning model and the ROC value is 0.694.

The sentences have been added in the last paragraph on page 10, “Besides, imaging modalities such as CT and MRI are recommended the first-line diagnostic methods to identify and predict different pathological states of HCC^{36,37}. For example, clinical staging of HCC is mainly diagnosed based on imaging characteristics, including the number and size of HCC nodules, and the presence of vascular invasion. Here, Raman

spectra have also shown feasibility for the determination of microvascular invasion with an accuracy of about 67% and the AUC value of 0.694 based on 84 patients (Supplementary Fig. 6). The results may be improved with a further increase in sample numbers and spectral collections.”

The following reference was added in References section:

36. Choi, J. Y., Lee, J. M. & Sirlin, C. B. CT and MR imaging diagnosis and staging of hepatocellular carcinoma: part II. Extracellular agents, hepatobiliary agents, and ancillary imaging features. *Radiology* 273, 30-50 (2014).

37. Jiang, H. Y. et al. Noninvasive imaging of hepatocellular carcinoma: From diagnosis to prognosis. *World J Gastroenterol* 24, 2348-2362 (2018).

2. On Page 10, the authors described tumor heterogeneity posed greater challenges to discrimination of different stages and differentiation grades of tumor tissues, with accuracies of 79.2% and 75.7% and to judgement of the liver cancer subtypes HCC and ICC, yielding an identification accuracy of 86.3%. Are these results of diagnostic accuracy better than traditional CT, MRI and US?

[Response]:

Thank you for pointing out this concern. First of all, unlike traditional imaging modalities, Raman spectroscopy is mainly employed for histopathological identification, especially the distinction between cancer tissue and adjacent tissue. Furthermore, we expect that the Raman spectra can reveal the alterations of biochemical components of pathological tissues, so as to identify tumour subtypes, cancer stages and differentiation levels. Noninvasive imaging approaches, such as dynamic multiphasic CT and MRI, are considered the first-line diagnostic modalities for HCC. A biopsy may be reserved in cases of indeterminate nodules that do not meet radiologic criteria for HCC.

HCC and ICC are the two most common primary liver cancers. It is essential to differentiate between HCC and ICC, which may affect the choice of treatment modality and subsequent treatment outcome. However, ICC may have a similar radiological appearance to HCC on CT and MRI, leading to misdiagnosis of ICC (*BMC Cancer*. 2019, 19, 1041). Recently, a study evaluated the association of specific pre-operative imaging characteristics in patients with histologically proven resected ICC (91 patients) (*World J. Surg.* 2020, 44, 3862-3867). The results showed that 24.3% of patients with ICC showed both two imaging characteristics of HCC (arterial phase hyperenhancement and non-peripheral venous washout) among those with no risk factors for HCC, while between 20.0 and 33.3% of patients with pure ICC fulfilled the imaging criteria for HCC among patients with risk factors for HCC. In another retrospective study, contrast-enhanced ultrasonography (CEUS) was used for the diagnosis of cirrhotic ICC patients. In 25 patients with small ICC, the misdiagnosis rate of HCC was 52% based on CEUS, while the misdiagnosis rates of CT (performed in 24 ICC) and MRI (11 ICC) were 4.2% and 9.1%, respectively (*Liver Int.* 2013, **33**, 771-779).

Generally, the prognosis of solid tumours is greatly affected by the tumour stage. According to different classification principles, several staging systems have been proposed, and different classification systems may get different staging results. Most current systems incorporate radiologic staging, referring to the imaging-based identification of the number and size of HCC nodules, the presence of vascular invasion, as well as extrahepatic metastases. Among them, Barcelona Clinic Liver Cancer (BCLC) classification is currently the most widely accepted in many countries as the standard staging system for HCC (*Hepatology* 2011, 53, 1020-1022). However, according to a study comparing long-term survival in 1713 prospectively enrolled HCC patients based on 5 different staging systems, BCLC may lack the ability to differentiate the prognosis in patients at a very early stage (*Cancer*, 2010, 116, 3006-3014).

The pathological grade is one of the factors influencing intrahepatic recurrence. High-grade HCC tumours usually have a higher intrahepatic recurrence rate.

Traditionally, a pathological examination is required to determine tumour grade, but there have been some reports using preoperative imaging analysis for grade prediction. For example, non-contrast-enhanced MRI-based radiomics signatures were used to preoperatively predict HCC grade. Radiomics signatures, including the extracted information of T1-weighted imaging (WI) and T2WI, could categorise high-grade and low-grade HCC cases (170 patients), with accuracies of 60-70% (*Eur. Radiol.* 2019, 29, 2802-2811). In another single-centre retrospective study, contrast-enhanced computed tomography (CECT)-based radiomics signatures were performed for preoperative prediction of pathological grades of HCC via machine learning. The accuracy was 53.33% for differentiating high-grade and low-grade HCC (297 subjects) (*Eur. Radiol.* 2020, 30, 6924-6932). Table R2-1 summarizes the diagnostic results of liver cancer with different pathological types based on imaging modalities.

In conclusion, in terms of accuracy, Raman spectroscopy is comparable to or even better than traditional imaging modalities in identifying different pathology types in the current study, although the two techniques have different application scenarios.

Table R2-1 Comparison of diagnostic results of liver cancer based on traditional imaging modalities and Raman spectroscopy in this study.

Pathological categories	Diagnostic results	References
HCC/ICC	misdiagnosis rate of HCC for ICC patients with no risk factors for HCC was 24.3% (36 ICC) by CT and MRI misdiagnosis rate of HCC for ICC patients with any risk factors for HCC was between 20.0 and 33.3% (54 ICC) by CT and MRI	World J. Surg. 2020, 44, 3862-3867
HCC/ICC	misdiagnosis rate of HCC was 52% (25 ICC) by CEUS, 4.2% (24 ICC) and 9.1% (11 ICC) by CT and MRI	Liver Int. 2013, 33 , 771-779
High-grade and low-grade HCC	accuracies are 60-70% (170 patients) based on non-contrast-enhanced MRI an accuracy of 53.33% (297	Eur. Radiol. 2019, 29, 2802-2811 Eur. Radiol. 2020, 30,

	subjects) based on contrast-enhanced CT	6924-6932
HCC	the sensitivities of US were 63-65%	Aliment. Pharmacol. Ther. 2009, 30, 37-47
HCC	the sensitivities of dynamic CT and MRI were 63%-76% and 77%-90%, and the specificities were 87%-98% and 84%-97%	World J. Gastroenterol. 2018, 24, 2348-2362
Per-lesion for nodular HCC	sensitivity of MRI for nodular HCC of all sizes is 77%–100%, while that of CT is 68%–91%	
	per-lesion sensitivities are 100% for both modalities for nodular HCCs larger than 2 cm, 44%–47% (MRI) and 40%– 44% (CT) for 1–2 cm HCCs	Radiology 2014, 273, 30-50, Hepatology 2003, 38, 1034-1042, and Gut 2010, 59, 638-644
	and 29%–43% (MRI) and 10%–33% (CT) for HCCs smaller than 1 cm	
Liver cancer / paracancer tissues	sensitivity 90.8% specificity 94.6% accuracy 92.6%	
HCC / ICC tissues	sensitivity 82.6% specificity 81.5% accuracy 82.4%	
Early / Advantaged stage	sensitivity 84.1% specificity 65.5% accuracy 78.3%	in this study
Well and moderate / poor grade	sensitivity 72.9% specificity 70.8% accuracy 72.3%	

[Action taken]:

A new table has been added in Supplementary information as Supplementary Table 4 (shown above as Table R2-1), as follows:

Supplementary Table 4. Comparison of diagnostic results of liver cancer based on traditional imaging modalities and Raman spectroscopy in this study.

Pathological categories	Diagnostic results	References
--------------------	------------

HCC/ICC	misdiagnosis rate of HCC for ICC patients with no risk factors for HCC was 24.3% (36 ICC) by CT and MRI misdiagnosis rate of HCC for ICC patients with any risk factors for HCC was between 20.0 and 33.3% (54 ICC) by CT and MRI	World J. Surg. 2020, 44, 3862-3867
HCC/ICC	misdiagnosis rate of HCC was 52% (25 ICC) by CEUS, 4.2% (24 ICC) and 9.1% (11 ICC) by CT and MRI	Liver Int. 2013, 33, 771-779
High-grade and low-grade HCC	accuracies are 60-70% (170 patients) based on non-contrast-enhanced MRI an accuracy of 53.33% (297 subjects) based on contrast-enhanced CT	Eur. Radiol. 2019, 29, 2802-2811 Eur. Radiol. 2020, 30, 6924-6932
HCC	the sensitivities of US were 63-65%	Aliment. Pharmacol. Ther. 2009, 30, 37-47
HCC	the sensitivities of dynamic CT and MRI were 63%-76% and 77%-90%, and the specificities were 87%-98% and 84%-97%	World J. Gastroenterol. 2018, 24, 2348-2362
Per-lesion for nodular HCC	sensitivity of MRI for nodular HCC of all sizes is 77%–100%, while that of CT is 68%–91% per-lesion sensitivities are 100% for both modalities for nodular HCCs larger than 2 cm, 44%–47% (MRI) and 40%– 44% (CT) for 1–2 cm HCCs and 29%–43% (MRI) and 10%–33% (CT) for HCCs smaller than 1 cm	Radiology 2014, 273, 30-50, Hepatology 2003, 38, 1034-1042, and Gut 2010, 59, 638-644
Liver cancer / paracancer tissues	sensitivity 90.8% specificity 94.6% accuracy 92.6%	
HCC / ICC tissues	sensitivity 82.6% specificity 81.5% accuracy 82.4%	
Early / Advantaged stage	sensitivity 84.1% specificity 65.5% accuracy 78.3%	in this study
Well and moderate / poor grade	sensitivity 72.9% specificity 70.8% accuracy 72.3%	

The sentences have been added on page 11, “In sum, in terms of accuracy, Raman spectroscopy is comparable to or even better than traditional imaging modalities (such as CT, MRI, and US) in identifying different pathology types in the current study (Supplementary Table 4), providing a powerful complement to existing pathology diagnostic techniques.”

3. On Page 10, the authors mentioned about the sensitivity of serological AFP. It is well known that AFP alone is not enough to diagnose with HCC, which is usually diagnosed by US, CT or MRI. Is Raman spectroscopy superior to these modalities?

[Response]:

Thank you for your comment. As discussed in the previous question, Raman spectroscopy and traditional imaging modalities have different applicability and advantages. Raman spectroscopy is primarily used for histopathological analysis, including for tissue sections, ex vivo tissue blocks, and in vivo detection. The collected Raman spectra could reveal the type and content of biochemical molecules in the tissue. The biochemical composition of tissue may change during carcinogenesis, which could be reflected in the spectra. The advantage of Raman detection is that it can be easily detected at anytime and anywhere with high molecular specificity, rapid test and low to moderate cost.

Imaging modalities are recommended for non-invasive diagnosis of patients before surgery. The diagnostic accuracy of different imaging modalities also varies greatly (Table R2-1). The sensitivity of US for the detection of small HCC is highly operator and patient-dependent. In expert settings, its sensitivity is about 80%, while in standard settings the sensitivity is closer to 65% (*Aliment. Pharmacol. Ther.* 2013, 38, 303-312). In addition, US may not be sensitive enough to detect early-stage HCCs, with a pooled sensitivity of 63% (*Aliment. Pharmacol. Ther.* 2009, 30, 37-47).

Characteristic imaging features of HCC include hyperenhancement in the hepatic arterial phase and washout appearance in the portal venous and/or delayed phases

relative to the background liver (*Hepatology* 2018, 67, 358-380). According to recent meta-analyses, the sensitivities of dynamic CT and MRI were 63%-76% and 77%-90%, respectively, and the specificities were 87%-98% and 84%-97%, respectively (*World J. Gastroenterol.* 2018, 24, 2348-2362). Besides, the diagnostic outcome of CT and MRI is largely dependent on tumour nodule size (*Radiology* 2014, 273, 30-50). The per-lesion sensitivity of MRI for nodular HCC of all sizes is 77%–100%, while that of CT is 68%–91% (*Radiology* 2014, 273, 30-50, *Hepatology* 2003, 38, 1034-1042, and *Gut* 2010, 59, 638-644). The per-lesion sensitivities, stratified by size, are 100% for both modalities for nodular HCCs larger than 2 cm, 44%-47% (MRI) and 40%-44% (CT) for 1-2 cm HCCs, and 29%-43% (MRI) and 10%-33% (CT) for HCCs smaller than 1 cm (*Radiology* 2014, 273, 30-50, *Hepatology* 2003, 38, 1034-1042, and *Gut* 2010, 59, 638-644).

In this study, the accuracy for discriminating liver cancer and adjacent non-tumour tissue is over 90%, which is comparable to or even better than that of imaging modalities. Although imaging methods are recommended as the first line for non-invasive preoperative diagnosis, histopathology is still the gold standard for the diagnosis of liver cancer. Here, Raman spectroscopic pathology provides a powerful supplement to traditional slice staining pathology analysis techniques. The advantage of Raman detection is that it can be easily detected at anytime and anywhere without complicated staining procedures, and it also has advantages of high molecular specificity, rapid test as well as low to moderate cost.

[Action taken]:

The sentences have been added in the last paragraph on page 10, “Besides, imaging modalities such as CT and MRI are recommended the first-line diagnostic methods to identify or predict different pathological states of HCC^{36,37}. For example, clinical staging of HCC is mainly diagnosed based on imaging characteristics, including the number and size of HCC nodules, and the presence of vascular invasion. Here, Raman spectra have also shown feasibility for the determination of microvascular invasion with an accuracy of about 67% and the AUC value of 0.694 based on 84 patients

(Supplementary Fig. 6). The results may be improved with a further increase in sample numbers and spectral collections. In sum, in terms of accuracy, Raman spectroscopy is comparable to or even better than traditional imaging modalities (such as CT, MRI, and US) in identifying different pathology types in the current study (Supplementary Table 4), providing a powerful complement to existing pathology diagnostic techniques.”

A new table has been added in Supplementary information as Supplementary Table 4 (as mentioned in the previous response).

4. In Figure 4, the authors showed cancer margin delineation. In HCC, the tumor capsule clearly distinguishes between cancerous and non-cancerous tissues. Boundaries were not smooth in Raman images like Fig. 4d and 4e. Can't you see the tumor capsule in Raman images?

[Response]:

Thank you for sharing the comment. To demonstrate that Raman imaging can delineate tumour boundaries, we selected two tissue block samples as examples, one with an obvious tumour boundary and one with a relatively inconspicuous boundary. In Fig. 4d and 4e, the distinction between cancerous and non-cancerous regions was successfully achieved by Raman imaging.

In HCC, there may be incomplete capsules and nonsmooth margins, especially in invasive tumours (*Clin. Imaging* 2021, 76, 77-82). Therefore, in such cases, the tumour boundary obtained by Raman spectroscopy may also be not smooth or the tumour capsule may not be observed. Moreover, the smoothness of the boundary in Raman images is related to the imaging scan resolution. It should be noted that the imaging range selected for this study was $50 \times 50 \mu\text{m}$, and the scanning interval was $2 \mu\text{m}$. In such a micro size, it may be difficult to acquire completely smooth boundary imaging, especially for the invasive tumour.

Raman spectroscopy has been shown to have the ability to image and identify different pathological tissue regions (*Sci. Transl. Med.* 2015, 7, 309 and *Nat. Commun.* 2020, 11, 6172). As shown in Fig. 5, Raman images of liver tissue sections with typical

morphologies were also obtained, including normal, cancerous, steatohepatitis, fibrotic, and connective tissues. Likewise, we believe fibrous capsule structures can be observed in Raman imaging of tissue sections as well.

[Action taken]:

The sentence has been inserted on page 16, “The tumour boundaries were not very smooth in the Raman imaging (Figs. 4d and 4e, upper panel), probably due to the thin tumour capsule and the small imaging area ($50 \times 50 \mu\text{m}$) with micron-scale scanning interval ($2 \mu\text{m}$).”

On page 16, the sentence “The second tissue showed a relatively poor cancer boundary in the displayed region, mixing with hepatic parenchyma that may indicate the presence of carcinoma infiltration, which was barely detectable in the brightfield images.” has been revised into “The second tissue showed a relatively poor border of cancer in the displayed area (Figs. 4d and 4e, bottom panel), intermingling with the liver parenchyma, probably due to the presence of cancerous infiltration, which was barely detectable in brightfield images.”

5. In Figure 6, the authors showed a preliminary result of usage of a handheld fibre-coupled Raman spectrometer. Do we need Raman spectrometer to observe tumors that can be seen by inspection or intraoperative ultrasound? Isn't it difficult to observe at the micro level because of camera shake and respiratory fluctuations?

[Response]:

Thank you for pointing out this concern. In this study, confocal Raman microscopy was used to visualize and examine tissue samples in ex vivo tissue sample testing, but for intraoperative applications, we employed a portable Raman device for spectral acquisition.

A handheld fibre-coupled Raman spectrometer was used in open surgery for intraoperative navigation. During surgical resection, surgeons could employ imaging

techniques such as preoperative imaging, intraoperative ultrasound, or rely on inspection to prejudge the lesion area. Then, spectra are collected near suspected lesions to obtain more accurate information about tumour boundaries or lesion regions. Raman spectroscopy may also be able to detect lesions that cannot be determined by ultrasound with higher resolution. It can further be used to measure the excised tissue to confirm whether there is residual cancer tissue (*Adv. Sci (Weinh)*. 2021, 8, 2002788).

During intraoperative detection, the probe is held above the tissue surface to acquire spectra through optical fibre, without additional camera devices. Although respiratory fluctuations may impair the spectral quality, given that a normal adult breathes about 12-20 breaths per minute and the integration time of a single spectrum acquisition is 0.2 s, respiratory fluctuations may not significantly affect spectral acquisition.

However, to better solve this problem, in our follow-up work, we are building an automatic robot system handling the optical fibre probe to automatically scan the patient tissue for intraoperative Raman mapping. The robot gripper holding the optical fibre probe can move point by point on the plane according to previously set commands. The system is equipped with a laser distance sensor, which helps to adjust the probe height in real-time according to the distance between the surface and the probe, so that the spectra could be acquired at a fixed distance throughout the detection process. In addition, the probe can be controlled/moved according to the regular breathing fluctuations to avoid the influence of potential breathing fluctuations. Furthermore, for liver tumour resection and large area scanning, millimetre-level tumour boundary resolution can already meet the clinical requirements. Therefore, we expect the follow-up work could help doctors achieve more accurate lesion identification.

[Action taken]:

The following sentences have been added on page 25, “Besides, the limited spectral collection points with handheld Raman probe may lead to the miss of lesions during surgery, while the respiratory fluctuations may affect the spectral quality even though the Raman integration time is more than 10-fold shorter than the respiratory period. We

expect that an intelligent robot collaboration system can be used to assist in the intraoperative Raman imaging to solve such issues in our follow-up work.”

6. Liver cancer often occurs deep in the liver. How deep can you observe in Raman spectroscopy?

[Response]:

Thank you for your comment. Traditional Raman technique can probe depths beyond the near-surface tissue layers, typically several hundred micrometres deep to mm (*ACS Cent. Sci.* 2016, 2, 12, 885–895; *Biomed. Opt. Express.* 2020, 11, 762–774), which satisfies the spectral acquisition of tissue sections as well as superficial cancer tissues. In this study, a common confocal Raman microscope was employed for in vitro Raman measurements. The detection limit of the confocal Raman was about 200 μm depth under the 532nm laser for liver tissue measurements (Figure R2-2). Beyond this range, the Raman signal was nearly undetectable. Therefore, at the moment, the Raman spectroscopy as an auxiliary technology is limited for the measurement of superficial tissues and suspected lesions within 200 μm depth for the intraoperative diagnosis.

However, with the advent of a range of specialist techniques based around spatially offset Raman spectroscopy (SORS), the Raman technology could non-invasively probe living tissue through depths up to 5 cm (*Chem. Soc. Rev.* 2021,50, 556-568). These techniques may be further incorporated into our system for the intraoperative diagnosis of deeper tissue.

Figure R2-2. **Raman spectra of liver cancer collected in different depth.** **a** Raman spectra were acquired in different depths of liver tissue block, with a depth interval of 25 μm for each acquisition from the surface (0 μm) to 300 μm depth. **b** Depth-resolved Raman intensity contour plot of liver tissue.

[Action taken]:

Supplementary Figure 11 (shown above as Figure R2-2) has been added to Supplementary information as follows.

Supplementary Figure 11. Raman spectra of liver cancer collected in different depth. **a** Raman spectra were acquired in different depths of liver tissue block, with a depth interval of 25 μm for each acquisition from the surface (0 μm) to 300 μm depth. **b** Depth-resolved Raman intensity contour plot of liver tissue.

The sentence has been added on page 19, “In this study, the maximum detection depth of liver tissue by confocal Raman spectroscopy under 532 nm laser was about 200 μm (Supplementary Fig. 11). For deeper tissue detection, the integration of spatially offset Raman spectroscopy (SORS) may be applied to achieve centimetre-level depth detection⁴⁹.”

The following reference was added in References section:

49.Nicolson, F., Kircher, M. F., Stone, N. & Matousek, P. Spatially offset Raman spectroscopy for biomedical applications. Chem. Soc. Rev. (2020).

7. On Page 22, the authors described that Ramanomics could distinguish different pathological tissues more cost-effectively. How much does it cost to observe in Raman spectroscopy?

[Response]:

Thank you for your comment. As far as we know, LC/MS-based metabolomic analysis typically requires complex sample preparation and testing workflows, costing \$100-\$300 for a single sample test. For traditional histopathological analysis, such as HE staining and immunohistochemical staining, additional staining reagents and related staining consumables need to be employed. However, for Raman spectroscopy measurement, most biological samples can be directly used for spectral testing without additional processing or other consumables. In our institute, confocal Raman microscopy tests cost around \$9 per hour. However, for future applications, excluding the machine cost, the daily cost of Raman measurements is merely the electricity.

Therefore, compared with the traditional metabolomic methods, Ramanomics is more cost-effective, with simple pretreatment, shorter detection time and without the requirement of additional consumables.

[Action taken]:

On page 23, the following sentence “The result was consistent with that of Raman analysis, demonstrating that Raman-based metabolomics, also known as Ramanomics⁵¹, could bring comprehensive and reliable biological information as traditional metabolomics, and distinguish different pathological tissues more conveniently and cost-effectively.” has been changed to “The result was consistent with that of Raman analysis, demonstrating that Raman-based metabolomics, also known as Ramanomics⁵¹, could bring comprehensive and reliable biological information as traditional metabolomics, and distinguish different pathological tissues more conveniently and cost-effectively without additional consumables.”

8. In Figure 2, legends of 2c and 2d are opposite. Fig. 2c is for tumor stage and Fig. 2d for tumor differentiation.

[Response]:

We apologize for the typo. As per the reviewer's suggestion, we have modified and included it in the revised manuscript.

[Action taken]:

The caption of Figure 2 has been revised as “**a-d** The average Raman spectra of 120 carcinoma and 120 paracarcinoma tissue samples (a), cancer tissue samples of patients with HCC and ICC (b), HCC tissue samples at different tumour stages (c), and HCC tissue samples with different cancer cell differentiation grades (d). The shaded areas represent the standard deviations of the means.”

9. “Tumour” and “tumor” are mixed in the article.

[Response]:

Thank you for your comments. We have checked and unified with “tumour” throughout the paper.

Reviewer #3 (Remarks to the Author): expert in deep learning network analysis

This paper proposed a new system for histopathological diagnosis of liver cancer based on Raman spectroscopy and deep learning. First, a CNN-based DL pipeline is used to classify the spectral data into paracancer or cancer tissues. The model is further used to differentiate different pathological types of liver cancer tissues. Last but not least, a portable Raman system was employed in surgery to validate this pipeline.

While the whole work is very novel and potentially impactful to bring benefits in liver cancer diagnosis, the study suffers from the following limitations:

1) The spectral data analysis. A liver tissue Raman database was established with 50 spectra per tissue sample, and a total of 12,000 spectra were obtained from 120 pairs of liver tissue samples. When training the VGG-model, the dataset is randomly splitted into 80% training, 10% validation and 10% test. However, due to the potential correlations of raman spectral data at the same tissue, a purely random split of training and testing data can cause label leakage, and therefore inflate the accuracy. A proper split would be following: take a random 100 pairs of liver tissue samples and all the raman spectral data on these tissue samples as training, and use the spectral data on the remaining 20 pairs of liver tissue sample as testing. This design will ensure the fair evaluation of DL-based models without potential label leakage. Further, the compared method, OPLS-DA is quite a weak comparison in my view. It would really demonstrate the power of DL-based approach by comparing with stronger baselines such as random forest, xgboost.

[Response]:

Thank you for the above constructive suggestions. We accordingly take your advice and use a random 100 pairs of liver tissue samples as the training and validation set (with a ratio of 8:2), and the remaining 20 pairs of liver tissue samples as the test set. The remaining three groups (HCC/ICC subtype, tumour stages, and differentiation grades) were also re-splitted (as discussed in the next question). The VGG model was

used for the re-evaluation of this new data set. As shown in Figure R3-1, in the differentiation of paracancer and cancer tissue, the diagnostic accuracy is 92.6%, which is consistent with the previous results (92.2%). The AUC value was 0.965, which is slightly lower than before (0.976).

Figure R3-1. i,j Cross-entropy loss (i) and accuracy (j) in iterative training of the CNN. Cross entropy represents the mean square error between the predicted value and the true value. **k** Binary confusion matrices for the classification of four tissue categories based on the CNN algorithm in percent (%). **l** ROC curves and corresponding AUC values.

As per the reviewer's suggestions, we have compared the power of the VGG model with other different baselines, including PLS-DA, random forest, and XGBoost. As listed in Table R3-1, compared with other common machine learning algorithms, the deep learning approach shows superior computational performance with higher accuracy in tissue identification of different pathological types. Especially, it has better performance in dealing with imbalanced data.

Table R3-1. Comparison of the capability of the VGG model and machine learning algorithms to identify different pathological types of liver tissue.

	Model	Accuracy (%)	Sensitivity (%)	Specificity (%)
Carcinoma	PLS-DA	84.8	95.2	73.3
	XGboost	87.9	91.8	83.5
	Random forest	86.4	87.5	85.2
	VGG-16	92.6	90.8	94.6
HCC	PLS-DA	76.9	93.5	10.5
	XGboost	77.8	86.8	42.0
	Random forest	77.0	79.0	69.0
	VGG-16	82.4	82.6	81.5
Advanced stage	PLS-DA	63.3	4.2	94.6
	XGboost	73.8	31.5	93.0
	Random forest	75.5	75.4	75.5
	VGG-16	78.3	65.8	84.1
Poor	PLS-DA	76.5	52.4	87.5
	XGboost	69.6	24.0	90.4
	Random forest	71.1	71.6	70.9
	VGG-16	72.3	70.8	72.9

[Action taken]:

Figure 2 has been updated as follows.

Fig. 2 Raman analysis of liver tissues and histopathological diagnosis based on deep learning. a-d The average Raman spectra of 120 carcinoma and 120 paracarcinoma tissue samples (a), cancer tissue samples of patients with HCC and ICC (b), HCC tissue samples at different tumour stages (c), and HCC tissue samples with different cancer cell differentiation grades (d). The shaded areas represent the standard deviations of the means. **e-f** Typical photographs of paracancer tissue (left) and liver

cancer tissue sample (right) (e) and the corresponding images of the H&E-stained tissues (f) tested in this study. **g** Raman testing of liver tissue with a micro-Raman spectrometer. **h** The architecture of the VGG-16-based deep learning model. Raman data consisting of 12,000 spectra were fed into the initial convolutional layer with 64 filters. Each convolutional layer had a kernel size of 3, connecting with a ReLU activation layer. A drop-out layer was utilized in full connection layers, following the basic blocks. Max-pooling (size 2, stride 2) was employed between blocks to reduce data length. The numbers below each block refer to the length and the number of channels of output respectively. **i,j** Cross-entropy loss (i) and accuracy (j) in iterative training of the CNN. Cross entropy represents the mean square error between the predicted value and the true value. **k** Binary confusion matrices for the classification of four tissue categories based on the CNN algorithm in percent (%). **l** ROC curves and corresponding AUC values.”

A new table has been added in supplementary information as Table 3 (shown above as Table R3-1), as follows.

Supplementary Table 3. Comparison of the capability of the VGG model and machine learning algorithms to identify different pathological types of liver tissue.

	Model	Accuracy (%)	Sensitivity (%)	Specificity (%)
Carcinoma	PLS-DA	84.8	95.2	73.3
	XGboost	87.9	91.8	83.5
	Random forest	86.4	87.5	85.2
	VGG-16	92.6	90.8	94.6
HCC	PLS-DA	76.9	93.5	10.5
	XGboost	77.8	86.8	42.0
	Random forest	77.0	79.0	69.0
	VGG-16	82.4	82.6	81.5
Advanced stage	PLS-DA	63.3	4.2	94.6
	XGboost	73.8	31.5	93.0
	Random forest	75.5	75.4	75.5
	VGG-16	78.3	65.8	84.1

Poor	PLS-DA	76.5	52.4	87.5
	XGboost	69.6	24.0	90.4
	Random forest	71.1	71.6	70.9
	VGG-16	72.3	70.8	72.9

On page 10, we changed “As a result, an accuracy of 92.2% was obtained for estimating carcinoma tissue area, accompanied by a sensitivity and specificity of 91.4% and 93.1%, respectively.” to “As a result, an accuracy of 92.6% was obtained for estimating carcinoma tissue area, accompanied by a sensitivity and specificity of 90.8% and 94.6%, respectively.”

On page 10, we changed “It is obvious that tumour heterogeneity posed greater challenges to discrimination of different stages and differentiation grades of tumour tissues, with accuracies of 79.2% and 75.7%, respectively. But a better result was acquired for judgment of the liver cancer subtypes HCC and ICC, yielding an identification accuracy of 86.3%. Moreover, 5-fold cross-validation (CV) was performed for each model to evaluate its generalizability. The CV results were consistent with the classification accuracy discussed above (Supplementary Table 3). In addition, four receiver operating characteristic (ROC) curves were plotted to quantitatively verify the performance of classifiers (Fig. 2l), which yielded good results with the area under the curve (AUC) ≥ 0.818 .” to “It is obvious that tumour heterogeneity posed greater challenges to discrimination of different stages and differentiation grades of tumour tissues, with accuracies of 78.3 and 72.3%, respectively. But a better result was acquired for judgment of the liver cancer subtypes HCC and ICC, yielding an identification accuracy of 82.4%. Four receiver operating characteristic (ROC) curves were plotted to quantitatively verify the performance of classifiers (Fig. 2l), which yielded good results with the area under the curve (AUC) between 0.783 and 0.965. Furthermore, compared with other common machine learning algorithms, including PLS-DA, random forest, and XGBoost, the deep learning approach shows superior computational performance with higher accuracy in tissue identification of different pathological types, especially in dealing with imbalanced data

(Supplementary Table 3).”

The sentences have been added in Methods on page 30, “Spectral data of 20 pairs of liver tissue samples were randomly selected as a test set, and spectral data of the remaining 100 pairs of samples were randomly divided into a training set and a validation set in a ratio of 8:2.”

2) The subsequent experiments of CNNs are conducted to classify multiple clinical phenotypes such as tumor stages, HCC/ICC subtypes. Similar to the concerns of potential label leakage mentioned above, the data splits need to be re-considered to justify the results. Further, to better understand the method, details about how to preprocessing the images, and how to augment data, how to alleviate the imbalance of labels should be discussed.

[Response]:

Thank you for pointing out the concern. To avoid potential label leakage, spectral data of different pathological type groups, including HCC/ICC subtype, tumour stages, and differentiation grades, were re-splitting. Specifically, about 20% of the samples in each group are randomly selected as the test set, and the spectral data of the remaining samples are divided into the training set and the validation set according to the ratio of 9:1.

The recalculated results are shown in Figure R3-1 k. The accuracy of CNNs-based identification of HCC/ICC subtype, tumour stages, and differentiation grades is 82.4%, 78.3%, and 72.3%, respectively, which is slightly lower than the results before the re-splitting (86.3%, 79.2%, and 75.7%). According to the model results, a good ability to distinguish different pathological tissues is still valid.

The CNN architecture was modified based on the VGG-16 framework, and the parameters and details of the model have been introduced in detail in the Methods section. The dataset of tumour subtype, tumour stage, and differentiation grade may suffer from data imbalance, so we employ weight balancing to address this issue.

The codes of the CNN algorithm have been updated and uploaded to GitHub (https://github.com/thidoiSanren/CNN_liver-cancer_Raman). The code for weight balancing as follows can also be found in the link.

```
weights = torch.tensor([num1, num2], dtype=torch.float32)
weights = weights / weights.sum()
weights = 1.0 / weights
weights = weights / weights.sum()
print(weights)
weights = torch.tensor(weights, dtype=torch.float32).to(device)
# The num1 is the number of group 0 and the num2 is the number of group 1.
```

Weight balancing balances the data by altering the weight of each training sample when computing the loss to make all classes contribute equally to the loss, resulting in more reliable results. In addition to accuracy, confusion matrix results are also used for model performance validation to reduce the effects of data imbalance.

As for the data acquisition and image processing of Raman imaging, we used a commercial software WIRE from Renishaw and its built-in image algorithm, without additional image preprocessing.

[Action taken]:

Figure 2 has been updated as shown in the previous response.

Paragraph 2 on page 10 has been revised into “In addition, three other CNN models were established to distinguish HCC from ICC tissues, and among tissues with different cancer stages and differentiation grades. The performance of four binary models is shown in the confusion matrices of Fig. 2k. It is obvious that tumour heterogeneity posed greater challenges to discrimination of different stages and differentiation grades of tumour tissues, with accuracies of 78.3 and 72.3%, respectively. But a better result was acquired for judgment of the liver cancer subtypes HCC and ICC, yielding an identification accuracy of 82.4%. Four receiver operating characteristic (ROC) curves were plotted to quantitatively verify the performance of classifiers (Fig. 2l), which

yielded good results with the area under the curve (AUC) between 0.783 and 0.965. Furthermore, compared with other common machine learning algorithms, including PLS-DA, random forest, and XGBoost, the deep learning approach shows superior computational performance with higher accuracy in tissue identification of different pathological types, especially in dealing with imbalanced data (Supplementary Table 3).”

The sentences have been added in Methods on page 30, “Similar splits were performed in other three classification models that 20% of the samples in each group are randomly selected as the test set, and the remaining samples are divided into the training set and the validation set according to the ratio of 9:1. Weight balancing was used to alleviate possible data imbalances by altering the weight of each training sample when computing the loss to make all classes contribute equally to the loss (detail of the code can be found in the following GitHub link).”

3) Tissue metabolomics analysis. The authors find a few metabolic features that can distinguish HCC and adjacent tissues. From Figure 3a, 3b, there are generally two types of metabolomics signatures: one is enriched in HCC and the other enriched in adjacent tissues. While the pattern is clear, the predictive power of these metabolomics is not explored. Further, it would be very interesting and informative to compare the predictive power of these metabolomic features with the deep learning models developed using the spectral data.

[Response]:

Thank you for sharing the comment. Non-targeted metabolomic analysis of 25 pairs of matched HCC tissues and adjacent non-tumour tissues was performed. We found significant differences in metabolic patterns between the two groups. A total of 108 characteristic differential metabolites were identified, most of which were downregulated in HCC tissues.

As per reviewer's comments, the metabolomic data were fed into the CNN model used for spectral analysis to distinguish HCC and tissues from adjacent non-tumour tissues. However, the diagnostic result is not stable enough due to the limited amount of data. We assign all the data of 25 pairs of samples to a training set and a validation set. When the ratio of the training set to the validation set is 7:3, the accuracy is 73.3%, while when the ratio is 8:2, the accuracy is 80%, both of which are lower than the results of spectral analysis.

In previous studies about the serum-based metabolic characterization of HCC, the diagnostic accuracy of HCC can reach 90% with sufficient samples (*Cancer Res.* 2013,73, 4992-5002). Therefore, we believe that the diagnostic accuracy based on metabolomic data can also be improved when more tissue samples are provided.

[Action taken]:

The sentences have been added on page 13, "Furthermore, the predictive power of the metabolomics was also investigated by the CNN model used for spectral analysis to distinguish HCC tissues from adjacent non-tumour tissues. The accuracy is between 70%-80%, which is lower than the spectral analysis results, but the accuracy may be improved as increasing the number of samples."

4) The intraoperative liver cancer diagnosis using a portable Raman system is extremely impressive! However, while the authors show different patterns of spectral Raman shift for different cancer tissues, the connection with previously developed DL models is missing. It is unclear whether the authors had validated the trained DL models using this portable Raman system, which would be extremely valuable to do so as a true validation of generalization.

[Response]:

Thank you for your valuable suggestion. In this study, a handheld portable Raman spectroscopy system was employed intraoperatively to detect hepatic carcinoma in six patients. To avoid interference with surgery, Raman signals were collected at several

randomly selected points in potential cancer and adjacent non-tumour areas in a short period. Therefore, the total amount of data collected by the portable Raman spectrometer is not as enough as the spectra collected in vitro.

In addition, Raman spectra of tissue collected by the portable Raman spectrometer are at variance with those obtained from the micro-Raman spectrometer, specifically at some peak positions. This may be attributed to differences between the two types of spectral equipment, such as laser sources, laser power, excitation wavelength, and spectrometer. Therefore, the CNN model trained by spectral data collected by the micro-Raman spectrometer may not be validated by the data collected by the portable Raman spectrometer.

However, based on statistical analysis, there are still significant differences in the spectral data of cancer tissue and adjacent tissue collected by portable Raman spectrometer. In the follow-up work, we will continue to collect more intraoperative spectral data and combine it with a robot arm for the Raman mapping of tissue. We believe that CNN-based intraoperative HCC diagnosis can still be achieved with sufficient spectral data.

[Action taken]:

The sentence in the first paragraph on page 23, “These spectral differences could help to distinguish tumour and hepatic parenchymal regions in surgery, and further intraoperative Raman mapping techniques may make it feasible to visualize tumour boundaries.” has been revised into “As more intraoperative spectral data are acquired, these spectral differences combined with suitable algorithms can help distinguish tumour and hepatic parenchymal regions in surgery, and further intraoperative Raman mapping techniques may make it feasible to visualize tumour boundaries.”

5) Transparency of reproducibility. The github link provided in the manuscript simply contains the codebase of plain VGG model training scripts. Without the spectral/imaging data, it is impossible for other researchers to validate the code or deploy the code. While I understand the privacy concerns of releasing patient data, I

would highly encourage the authors to at least provide de-identified or simulated data so that other researchers can test/validate the proposed methods.

[Response]:

Thank you for your comments. Actually, we submitted spectral data to the editor through the delivery system at our first submission. But we are sorry for not uploading it on GitHub. We accordingly updated the demo dataset on the GitHub link (https://github.com/thidoiSanren/CNN_liver-cancer_Raman).

[Action taken]:

On page 31 section “Code availability”, we modified the sentence as follows “The code used for the CNN model and demo datasets are available at https://github.com/thidoiSanren/CNN_liver-cancer_Raman”.

Reviewer #4 (Remarks to the Author): expert in metabolomics

The manuscript by Huang et al, describes a methodology to use Raman spectroscopy and deep learning to identify liver cancer tissue. The goal is to apply this in the surgery theatre for intraoperative pathological diagnosis (compared to classical biopsy methods) to better resect cancer tissue.

The analyses were also compared to metabolomics analyses.

Focusing mainly on the Metabolomics analysis, the performed overall analysis looks robust. I have two comments:

1. The data representation in e.g. panel 3d appears not very intuitive. I am wondering if a fold change representation of HCC sample over respective adjacent tissue would be more intuitive (in such case, the signal for each metabolite from adjacent tissue would always be = 1).

[Response]:

Thank you for your valuable suggestion. To make the data in Fig. 3d more intuitive, we have added a new figure as Supplementary Figure 7 (shown as in Figure R4-1) in supplementary information, with a fold change representation of the HCC sample over corresponding adjacent tissue. It is evident from Supplementary Figure 7 that phosphatidylcholines (PCs) tagged with polyunsaturated fatty acids (PUFAs) were down-regulated in HCC tissues, while an increase of PCs tagged with saturated fatty acids (SFAs) and monounsaturated fatty acids (MUFAs) in HCC group was observed. Besides, most nucleosides, bases, and saccharides were down-regulated in the HCC group, except inosine, 1-methyl-hypoxanthine, guanosine, and glucose 6-phosphate.

Figure R4-1. Relative contents of representative phosphatidylcholines (a-b), nucleosides, bases (c), and saccharides (d) with significant differences between HCC tissues and adjacent tissues. The signal for each metabolite from adjacent tissue was set to 1. Error bars show the standard deviations. HCC tissues, n=25, adjacent tissues, n=25.

[Action taken]:

A new figure Supplementary Figure 7 (shown above as Figure R4-1) has been added as follows,

Supplementary Figure 7. Relative contents of representative phosphatidylcholines (a-b), nucleosides, bases (c), and saccharides (d) with significant differences between HCC tissues and adjacent tissues. The signal for each metabolite from adjacent tissue was set to 1. Error bars show the standard deviations. HCC tissues, n=25, adjacent tissues, n=25.

A new sentence has been added to the last paragraph on page 13, “To compare the above metabolite differences more intuitively, relative changes representation of HCC samples over respective adjacent tissues were shown in Supplementary Fig. 7.”

2. The analytical method is optimally suited for polar metabolites while lipids etc are barely retained by the used chromatographic method (all hydrophobic compounds elute at very early retention time). I am wondering if these compounds can be detected with confidence when barely separated? What are the measures applied to allow compound identification?

[Response]:

Thank you for pointing out this concern. In this study, LC-MS/MS analyses were performed based on a UHPLC system with a UPLC BEH Amide column coupled to a Q Exactive HFX mass spectrometer (as described in Methods). The analytical method is suited for polar metabolites, but this does not mean that it does not apply to hydrophobic compounds. The retention time of lipids is within the first 4 minutes (the first third of the total retention time), which is sufficient for them to be detected and identified. Supplementary Table 5 lists the retention times of some lipids such as PC (22:5/16:1), PC (16:0/14:0), and PC (15:0/15:0), with the retention time of 117.30 s, 152.18 s, 175.12 s, respectively. In addition to the lipids in the table, the retention times of other lipids such as LysoPE (16:0/0:0) and LysoPC (16:0) were at 226 s and 216 s, respectively.

The identification of compounds is based on a self-written R package and a self-built secondary mass spectrometry database (from Shanghai Biotree Biotech CO., LTD.), as described in the Method section on pages 28-29. A typical qualitative analysis process is as follows. Firstly, the molecular weight of the metabolites is determined according to the mass-to-charge ratio of the parent ions in the primary mass spectrometry. Then, secondary identification of metabolites is achieved based on the mass-to-charge ratio of characteristic product ions generated by the fragmentation of parent ions and the response intensity of product ions. Finally, a matching degree of the candidate metabolite is calculated and scored to characterize the analyte.

[Action taken]:

On page 29, the sentence “Then an in-house MS2 database (BiotreeDB) was used for annotating metabolites. The cutoff for annotation was set at 0.6.” was modified to “The identification of compounds was based on the mass-to-charge ratio of the parent ions in the primary mass spectrometry and characteristic product ions generated by the fragmentation. An in-house MS2 database (BiotreeDB) was used for annotating metabolites. The cutoff for annotation was set at 0.6.”

REVIEWERS' COMMENTS

Reviewer #1 (Remarks to the Author):

The authors have addressed my earlier comments in their revision. I now recommend its publication in Nature Comm.

Reviewer #2 (Remarks to the Author):

Thank you for your revisions. Tissue analysis by Raman spectroscopy has the potential to produce new knowledge, but now it is not superior to pathology, nor is it useful for intraoperative diagnosis.

Reviewer #4 (Remarks to the Author):

Thank you for replying to the comments provided. All my concerns have been addressed. I have no further remarks.

Reviewer #5 (Remarks to the Author):

The authors describe a method to identify primary hepatic malignant lesions (HCC and ICC) using a combination of Raman spectroscopy and AI-derived algorithms. The authors trained their model on 120 matched tumor-normal human tissue and reached a sensitivity and specificity of over 90% in differentiating tumor and normal liver tissue and also managed to define tumor grading and vascular invasion at lower percentages. Their method is rapid, non-disruptive, and label-free compared to conventional histology.

The following points need to be addressed:

- The authors provided detailed information about the clinical and histopathological diagnoses in supplementary table 1. The table is missing the other two central causes for cirrhosis and HCC- hepatitis C virus and alcohol consumption.
- Does the proposed methodology manage to differentiate HCC precursors from HCC (i.e large cell dysplasia and small cell dysplasia)? These precancer changes can't be differentiated grossly from HCC and require tissue biopsy and microscopic evaluation. HCC precursors have different management than HCC.
- Does the proposed algorithm manage to correctly identify non-malignant hepatic lesions that can be hard to identify on imaging (hepatocellular adenoma, focal nodular hyperplasia)
- The most common neoplastic liver lesions are not primary, but rather metastatic. Can the proposed methodology differentiate between primary hepatic neoplasms and secondary lesions? This has a significant impact on treatment and prognosis.
- The authors mention an optional usage of the proposed methodology to detect tumor margin in HCC. However, the current treatment guidelines in HCC require wide resection (partial hepatectomy in localized mass, complete hepatectomy following liver transplantation in cases where several HCC foci are detected). Hence, this margin assessment of HCC has no clinical relevance. However, in hepatic metastases resection (a far more common clinical scenario) this can be very relevant.
- In multiple places in the manuscript the term "carcinoma" or "hepatic carcinoma" is written- Do you mean either HCC or ICC?
- Are the sensitivity and specificity of Raman spectroscopy to differentiate between HCC and its surrounding tissue grade related? Well-differentiated HCC has a very similar morphology to the normal liver parenchyma.
- Line 164- "Clinically, over 90% of liver tumours are primary liver cancer..." This sentence is incorrect, as metastases are the most common cause of liver tumors. Change to primary liver

tumors or some equivalent term.

Reviewer #5 (Remarks to the Author):

The authors describe a method to identify primary hepatic malignant lesions (HCC and ICC) using a combination of Raman spectroscopy and AI-derived algorithms. The authors trained their model on 120 matched tumor-normal human tissue and reached a sensitivity and specificity of over 90% in differentiating tumor and normal liver tissue and also managed to define tumor grading and vascular invasion at lower percentages. Their method is rapid, non-disruptive, and label-free compared to conventional histology.

The following points need to be addressed:

1. The authors provided detailed information about the clinical and histopathological diagnoses in supplementary table 1. The table is missing the other two central causes for cirrhosis and HCC- hepatitis C virus and alcohol consumption.

[Response]

Thank you for your comments. The prevalence of risk factors for liver cancer is different across different regions, for example, in China, the endemic hepatitis B virus (HBV) and the development of chronic hepatitis B infection have been the main driver of liver cancer, whereas in South Korea and Japan, liver cancer is mainly caused by hepatitis C virus (HCV) infection (*J. Hepatol.* 2019, 70, 674-683).

The tissue samples collected in this study were all from patients without HCV infection, which was included in supplementary table 1. The information of alcohol consumption was also added in supplementary table 1.

[Action taken]:

Supplementary Table 1 has been modified as follows.

Supplementary Table 1. Clinical characteristics of the tested patients.

	Hepatocellular Carcinoma (HCC) (n=98)	Intrahepatic Cholangiocarcinoma (ICC) (n=22)
Age (yrs)	58.1±12.2	60.8±13.1
Males	84 (85.7%)	13 (59.1%)
Females	14 (14.3%)	9 (40.9%)
Alcohol consumption ^a		
no drinking	59 (60.2%)	16 (72.7%)
< 28 g/day	19 (19.4%)	2 (9.1%)
≥28 g/day	20 (20.4%)	4 (18.2%)
Hepatitis B Virus (HBV)	64 (65.3%)	5 (22.7%)
Hepatitis C Virus (HBV)	0	0
Hepatic Cirrhosis	72 (73.5%)	5 (22.7%)
Steatohepatitis	20 (20.4%)	6 (27.3%)
Stage	Early: 69 Advantage: 29	
Differentiation	Well: 27 Moderately: 40 Poorly: 31	
Alpha-Fetoprotein (AFP) (ng/mL)	Ave: 2939.2±10231.7 Median: 16.3	Ave: 56.9±179.0 Median:3.20

^a refers to the “Dietary Guidelines for Americans 2020-2025”

2. Does the proposed methodology manage to differentiate HCC precursors from HCC (i.e large cell dysplasia and small cell dysplasia)? These precancer changes can’t be differentiated grossly from HCC and require tissue biopsy and microscopic evaluation. HCC precursors have different management than HCC.

Does the proposed algorithm manage to correctly identify non-malignant hepatic lesions that can be hard to identify on imaging (hepatocellular adenoma, focal nodular hyperplasia)

[Response]

Thank you for pointing out these concerns. Actually, Raman spectroscopy has been successfully employed to differentiate cancer tissues from precursors, such as for the distinction of gastric carcinoma and early precursors (including intestinal metaplasia and dysplasia) (*J. Biophotonics* 2013, 6, 49-59, *Anal. Bioanal. Chem.* 2015, 407, 8303-10), and the identification of precursor of endometrial cancer (complex atypical hyperplasia) (*Sci. Rep.* 2021, 11, 9079). In addition, the differentiation of colorectal carcinoma and adenomatous polyps has also been reported (*Diagnostics (Basel)* 2021, 11, 2048).

Therefore, we believe the proposed methodology based on Raman spectroscopy and deep learning is applicable for the identification of HCC precursors and non-malignant hepatic lesions based on progressive changes of biochemical constituents in tissues. Although HCC precursor tissues and non-malignant liver lesions were not enrolled in this study, we would be interested in validating the ability of the method to identify these tissues in our following work.

[Action taken]:

The sentence has been added in the first paragraph on page 24, “Furthermore, we also expect to achieve the discrimination of HCC precursors and non-malignant liver lesions in the follow-up work, as well as the distinction between primary and secondary liver cancer, which is crucial to the treatment and prognosis of hepatic carcinoma.”

3. The most common neoplastic liver lesions are not primary, but rather metastatic. Can the proposed methodology differentiate between primary hepatic neoplasms and secondary lesions? This has a significant impact on treatment and prognosis.

[Response]

Thank you for sharing the comment. Because of the high incidence of hepatitis B,

the number of cases of primary hepatic carcinoma (PHC) and metastatic hepatic carcinoma (MHC) is roughly equal in China. In addition, some patients may have unresectable MHC at diagnosis. Here we only enrolled intraoperatively resected liver tissue samples from patients with PHC in this study. And we believe that the proposed method has the potential to distinguish primary hepatic neoplasms and secondary lesions as there are significant differences in biochemical composition between tissues and even the blood of patients (*Int. J. Mol. Sci.* 2018, 19, 3014, *Biomed. Res. Int.* 2022, 2022, 3541403).

[Action taken]:

The sentence has been added in the first paragraph on page 24, “Furthermore, we also expect to achieve the discrimination of HCC precursors and non-malignant liver lesions in the follow-up work, as well as the distinction between primary and secondary liver cancer, which is crucial to the treatment and prognosis of hepatic carcinoma.”

4. The authors mention an optional usage of the proposed methodology to detect tumor margin in HCC. However, the current treatment guidelines in HCC require wide resection (partial hepatectomy in localized mass, complete hepatectomy following liver transplantation in cases where several HCC foci are detected). Hence, this margin assessment of HCC has no clinical relevance. However, in hepatic metastases resection (a far more common clinical scenario) this can be very relevant.

[Response]

Thank you for your suggestion. Here we propose a spectral detection/imaging method, which is expected to be used in the judgment of tumour boundaries during surgery, and the identification of small residual lesions to determine whether the tumour is completely resected. As you mentioned, the identification of the margin may be more applicable to the resection of liver metastases clinically. And we have modified the relevant expressions as follows.

[Action taken]:

On page 24, the sentence “However, complete removal of cancer tissue with minimal damage to adjacent normal tissue has always been a challenge for surgeons who have lacked suitable intraoperative methods for visually distinguishing the two tissues.” has been revised into, “Accurate tumour boundary identification is helpful for complete resection of lesions without excessive resection of normal tissue, especially for hepatic metastases resection. However, this is often a challenge for surgeons who lack suitable intraoperative methods for visually distinguishing the two tissues.”

5. In multiple places in the manuscript the term “carcinoma” or “hepatic carcinoma” is written- Do you mean either HCC or ICC?

[Response]

Thanks for your comments. The term "carcinoma" or " hepatic carcinoma " is in opposition to "adjacent non-tumour tissue" and it refers to the primary liver cancer, i.e. HCC or ICC in the manuscript. For instance, in Figure 2, "carcinoma" refers to all cancerous tissue samples, including HCC and ICC tissues.

6. Are the sensitivity and specificity of Raman spectroscopy to differentiate between HCC and its surrounding tissue grade related? Well-differentiated HCC has a very similar morphology to the normal liver parenchyma.

[Response]

Yes, we suppose the distinction between HCC and its surrounding tissue is related to the differentiated grade. The greater the difference in biochemical composition between tissue samples, the more it helps us to distinguish between different types of tissues by Raman spectroscopy. As shown in Supplementary Figure 4, the overall spectral difference between moderately and poorly differentiated groups was more significant

than that of well- and moderately differentiated groups. Therefore, it is easier for us to achieve the distinction between moderately and poorly differentiated groups.

7. Line 164- “Clinically, over 90% of liver tumours are primary liver cancer...” This sentence is incorrect, as metastases are the most common cause of liver tumors. Change to primary liver tumors or some equivalent term.

[Response]

Thank you for your valuable suggestion. We have corrected the expression as follows.

[Action taken]:

On page 7, The sentence “Clinically, over 90% of liver tumours are primary liver cancer, of which 75–85% and 10–15% are hepatocellular carcinoma (HCC) and intrahepatic cholangiocarcinoma (ICC), respectively³⁴.” has been revised into, “Primary liver cancer is one of the most common cancers worldwide, of which 75–85% and 10–15% are hepatocellular carcinoma (HCC) and intrahepatic cholangiocarcinoma (ICC), respectively³⁴.”